# Keypoint-based modeling reveals fine-grained body pose tuning in superior temporal sulcus neurons

Rajani Raman [1,2,4], Anna Bognár[1,2,4], Ghazaleh Ghamkhari Nejad[1,2], Albert Mukovskiy[3], Lucas Martini [3], Martin Giese [3] & Rufin Vogels [1,2] ✉

Body pose and orientation serve as vital visual signals in primate non-verbal social communication. Leveraging deep learning algorithms that extract body poses from videos of behaving monkeys, applied to a monkey avatar, we investigated neural tuning for pose and viewpoint, targeting fMRI-defined mid and anterior Superior Temporal Sulcus (STS) body patches. We modeled the pose and viewpoint selectivity of the units with keypoint-based principal component regression with cross-validation and applied model inversion as a key approach to identify effective body parts and views. Mid STS units were effectively modeled using view-dependent 2D keypoint representations, revealing that their responses were driven by specific body parts that differed among neurons. Some anterior STS units exhibited better predictive performances with a view-dependent 3D model. On average, anterior STS units were better fitted by a keypoint-based model incorporating mirror-symmetric viewpoint tuning than by view-dependent 2D and 3D keypoint models. However, in both regions, a view-independent keypoint model resulted in worse predictive performance. This keypoint-based approach provides insights into how the primate visual system encodes socially relevant body cues, deepening our understanding of body pose representation in the STS.

Body pose is an important visual signal of non-verbal communication and emotional expression in primates. Primates employ body poses to signal e.g., submission or dominance to conspecifics. In addition, body orientation, e.g. facing away versus frontal, is an important visual cue in social behavior. Given their ethological relevance, both body view and pose, along with their interaction, are expected to be analyzed by the visual system. Indeed, single-unit studies in the macaque Superior Temporal Sulcus (STS) showed body responses that depended on viewpoint[1–4] and demonstrated selectivity for pose[1,4]. However, these studies employed only a limited set of poses (e.g. five in ref. 4) and therefore could not reveal the visual features underlying the body pose and viewpoint tuning of the STS neurons.

Here, we leveraged deep learning computer vision algorithms that extract the body pose from markerless motion capture of behaving monkeys[5] to understand the encoding of pose and viewpoint at the unit level. The captured poses were parameterized using 3D coordinates of joints and body features ("keypoints"), which were then transferred to a monkey avatar. The use of the avatar enabled the rendering of body poses from different viewing angles, resulting in a large variety of images of the monkey avatar displaying different poses seen from various vantage points. Using this stimulus set of poses, we performed a quantitative, model-based characterization of the pose and view selectivity of macaque STS neurons.

fMRI studies in monkeys revealed body-category selective regions (body patches) in their ventral STS[6–12]. These contain a high proportion

[1]Department of Neurosciences, KU Leuven, Leuven, Belgium. [2]Leuven Brain Institute, KU Leuven, Leuven, Belgium. [3]Section Computational Sensomotorics, Department N3, Hertie Institute for Clinical Brain Research & Centre for Integrative Neuroience, University Clinic Tübingen, Tübingen, Germany. [4]These authors contributed equally: Rajani Raman, Anna Bognár. ✉e-mail: Rufin.vogels@kuleuven.be

of body-selective neurons[4,13] (for review, see ref. [14]). We targeted the middle STS body patch (MSB) and the anterior STS body patch (ASB) for electrophysiological recordings, thus increasing the yield of body-selective neurons. This allowed a comparison of the pose and viewpoint selectivity between two hierarchical levels of the body processing network[15].

We modeled the body pose and viewpoint selectivity of each recorded STS unit using view-dependent 2D and 3D keypoint representations. Since we employed 2D stimuli, the depth dimension of the 3D representation requires inference. The modeling approach allowed us not only to quantify the selectivity for body pose and viewpoint but also to determine which keypoints, i.e., body parts, contribute to this selectivity. We show that the pose and viewpoint selectivity of mid STS units can be well-modeled by 2D keypoint representations. Specifically, this selectivity was captured by an axis in a space, in which the dimensions correspond to the principal components derived from 2D keypoint coordinates of natural poses and viewpoint data. Visualizing which keypoints were contributing to the selectivity showed that the response along this axis was typically driven by the location of a subset of body parts (e.g. the arms and tail/lower back), which varied among units. The selectivity of the anterior STS units was fitted to a lesser extent by the 2D keypoint model, but a view-dependent 3D model that considered the depth of the keypoints improved the predictive performance for some units. Incorporating mirror-symmetric view tuning in the keypoint model increased on average the fits of the anterior but not mid STS selectivity. View-dependent models provided on average a better predictive performance than a pure view-invariant 3D keypoint model at each level.

## Results

We measured the selectivity of macaque STS units to a set of 720 views and poses of a monkey avatar (Fig. 1a, b). The avatar, with a blurred face, displayed natural poses of rhesus monkeys derived from published video data[5] (Methods). The images, having a maximal extent of 6°, were presented centrally during passive fixation while recording spikes using V-probes from a mid STS region (MSB) and a more anterior region in the ventral bank of the STS (ASB) of two monkeys. These regions were selected based on a fMRI body patch localizer performed in the same monkeys (Fig. 1c). We included only units that were selective for the avatar stimuli (Methods) with a minimum reliability (Spearman-Brown corrected split-half correlation r) of 0.5 (median r: MSB: G = 0.72; T = 0.71; ASB: G = 0.74; T = 0.76). We selected units recorded from channels that were body-selective assessed by measuring the responses to videos of acting monkeys, dynamic faces, and dynamic objects[12,16] (Body Selectivity index > 0.33; Methods).

We parameterized the body pose of each image using the x, y, and z coordinates of 22 keypoints (Fig. 1d; Supplementary Fig. S1). The keypoint representation is fairly abstract since it reduces the body to its skeleton and ignores the texture, shading, and shape of local body parts. However, information about the relative positions of the keypoints of the body in the image is kept, reflecting the body pose in the different views. Initially, we employed two keypoint representations, a 2D and a 3D view-dependent (3D_VD) one. In the 2D representation, each keypoint consists of an x and y coordinate, reflecting the projection of the 3D keypoint on the image plane, whereas in the 3D_VD representation, each keypoint has x, y, and z coordinates, with z indicating the relative distance from the frontal plane. The z-coordinate is not explicitly present in the 2D stimulus and must be inferred.

First, we assessed whether the keypoint representations could model the units' selectivity for the body images. Therefore, we first performed a Principal Component Analysis (PCA) on the keypoint data of all 720 images for 2D and 3D_VD separately, thus capturing the covariance of the keypoint coordinates across pose and viewpoint (Fig. 1d). For the 2D and 3D representation, ten principal components

(PCs; Supplementary Figs. S2b, S3, S4b) explained close to 90% of the variance (Supplementary Figs. S2a, S4a). The ten PCs were employed to model the responses of each unit, using cross-validated multiple linear regression (Methods). For each unit, we quantified the predictive performance of a model by the coefficient of determination, $R^2$, computed for the left-out stimuli, normalized by the reliability of the unit ($\widetilde{R}^2$; Methods). This allows comparisons of model fits between units and regions, accounting for differences in the reliability of the selectivity. For the vast majority of units, $\widetilde{R}^2$ was significantly greater than that obtained when permuting stimulus labels for both regions, monkeys, and models (Fig. 2a; Methods). However, we observed differences between regions and models (Fig. 2a). For the 2D model, the median $\widetilde{R}^2$ was 0.41 and 0.44 for MSB units of G ($n = 147$) and T ($n = 138$), respectively, while significantly less for ASB units (G: 0.18 ($n = 197$; Wilcoxon Rank Sum test: $p = 8.0e-28$); T: 0.24 ($n = 108$; $p = 1.440e-12$)). For the 3D_VD model, the $\widetilde{R}^2$ were similar to those for 2D in MSB of both monkeys (medians: G: 0.39; T: 0.44), but higher in ASB (G: 0.22; T: 0.30; Wilcoxon signed rank tests: G: $p = 8.767e-21$; T: 2.662e-10), but MSB had still better fits than ASB for the 3D_VD model (G: $p = 1.233e-16$; T: $p = 3.729e-09$). The difference between the 2D and 3D model fits in ASB was present also when increasing the number of selected PCs (Supplementary Fig. S5).

As a benchmark to evaluate the performance of these keypoints models, we employed Convolutional Neural Networks (CNNs), which have become the standard models of IT neuronal selectivity[17,18]. We presented the images to AlexNet[19] and ResNet50_rbst[20] (Methods). We performed a PCA on the activations of each layer (Supplementary Fig. S6a) and employed the first 50 PCs to fit the responses of each unit using the same cross-validation regression procedure as for the keypoint models. Using more than 50 PCs did not increase the predictive performance substantially (Supplementary Figs. S6d, S7). In both regions, AlexNet layer 5 produced the best fit, with a higher $\widetilde{R}^2$ for MSB than ASB (Supplementary Fig. S6c). However, ASB showed a better fit than MSB for layer 7, as MSB showed a drop in fit for fully connected layers, despite those layers having higher cumulative explained variance for the 50 PCs than the convolutional layers (Supplementary Fig. S6). This aligns with ASB being at a higher hierarchical level than MSB. ResNet50_rbst produced better predictive performances than AlexNet, with the highest median fits for stages 4 and 5 for MSB and ASB, respectively, (Supplementary Fig. S6). For all ResNet50_rbst stages, the MSB fits were higher than those for ASB, which agrees with the keypoint models.

To compare the fits of the keypoint models and the CNNs, we computed the reliability-corrected adjusted $R^2$, depicted as $\widetilde{\widetilde{R}}^2$, controlling for the difference in the number of predictors between the models (Fig. 2b; Supplementary Fig. S6e; unadjusted $R^2$ data in Supplementary Fig. S7). For the convolutional layers 3 and higher AlexNet performed similar or significantly better than the keypoint models, whereas the fully connected layers performed similar or significantly worse (Wilcoxon signed rank tests; Fig. 2b, Supplementary Fig. S7). The keypoint models outperformed ResNet50_rbst for stages 1 to 3 for MSB, but the opposite was the case for stages 4 and 5 (Supplementary Figs. S6e, S7). For ASB units, the deeper layers of both CNNs performed significantly better than the keypoint models, especially the 2D model (Fig. 2b, Supplementary Figs. S6e, S7). These data show that deeper convolutional CNN layers can outperform the keypoint models, but the increase in predictive performance is rather modest (Fig. 2b, Supplementary Figs. S6e, S7). The keypoint model significantly outperformed layer 1 of the CNNs (Fig. 2b, Supplementary Figs. S6e, S7) and a pixel-based model (Methods; Supplementary Figs. S7, S8), showing that the keypoint model fits are not explainable by putative small receptive fields detecting stimulus parts but instead suggest a feature selectivity. Variance partitioning analysis (Methods) revealed that most explained variance was shared between CNNs and keypoint

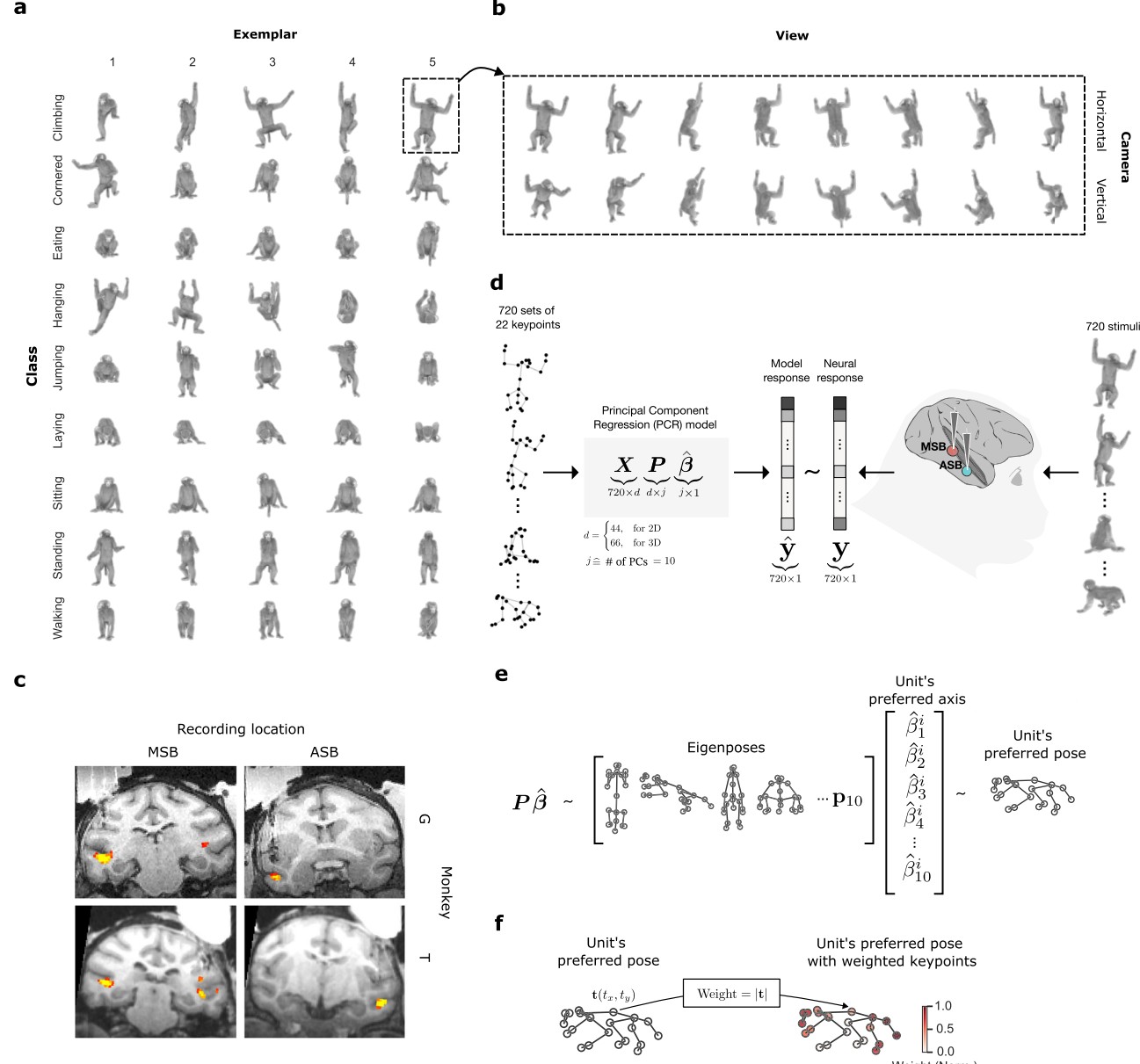

**Fig. 1 | Stimuli, recording locations, and model. a, b** The monkey pose stimulus set (*n* = 720 stimuli) includes a monkey avatar displaying various natural poses viewed from different vantage points. **a** 45 poses categorized into 9 action classes with 5 exemplars each. **b** 16 views for one example pose. The 16 views consisted of 8 orientations (45° steps of azimuth) of a horizontally positioned camera and 8 orientations (45° steps) rendered from a 45° camera elevation angle. **c** Targeted body patches MSB and ASB (yellow/red; in columns) of monkeys G and T (in rows). fMRI activations are obtained from the contrast bodies minus faces and objects. Artefacts from penetrations can be seen (black). **d** Logic of the keypoint-based modeling. We recorded the response of units to the pose stimulus set. The responses of a unit to the poses and viewpoints were fitted using a principal component regression (PCR) model. The model input consisted of a set of 22-keypoints (2D or 3D) extracted from the 720 stimuli, which were transformed into a data matrix $X \in R^{720 \times d}$, where $d = 44$ (22 keypoints x 2 (x, y) coordinates), or 66 (22 keypoints x 3 (x, y, z) coordinates). Principal components, P, of this data matrix were used as predictors in a cross-validated regression model to predict the neural response. The regression coefficients $\hat{\beta}$ represent the unit's "preferred axis" in the PC space. **e** Visualization of the preferred pose-viewpoint combination of a unit by inversion of the model. Transformation of $\hat{\beta} \in R^{10}$ to $P\hat{\beta} \in R^d$ was used to visualize the preferred pose-viewpoint combination. It can be interpreted as a weighted sum of 'Eigenposes' derived from the PCs, with the weights determined by the beta values associated with those components. **f** We estimated the weight (contribution) of each keypoint, indicated by color, by calculating the magnitude of the (x, y) or (x, y, z) coordinates of the transformed axis **t** (Methods) for each keypoint.

models (Supplementary Fig. S9), as expected since CNNs can encode image features related to pose and viewpoint differences. Keypoint models showed a small amount of unique explained variance, likely capturing shape differences not accounted for by the texture-biased CNNs[21]. CNNs explained more unique variance, probably reflecting their sensitivity to texture/shading cues absent in the keypoint models. CNNs showed higher relative unique variance in ASB than MSB, possibly reflecting a greater reliance on texture/shading in ASB. This aligns

with mid STS prioritizing face shape, while anterior IT favors facial "appearance"[22].

Although the predictive performances of the keypoint models were somewhat lower than those obtained with the CNNs, the former models, with only linear transformations (PCA and linear regression), have the advantage that we can invert them (Fig. 1e, f) and visualize the keypoints that drive the pose and viewpoint selectivity (Methods). The model of each unit defines a "preferred axis" in the PC space which

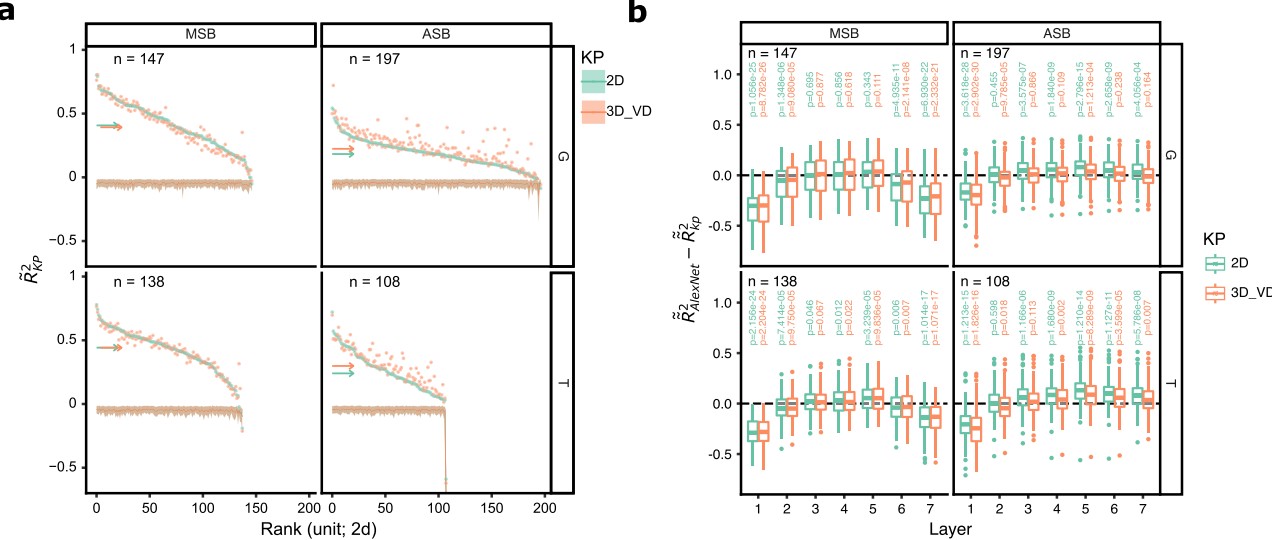

**Fig. 2 | Model performance. a** Reliability-normalized coefficient of determinations $\tilde{R}^2_{KP}$ for the units for the 2D and 3D_VD keypoint models. Units were ranked according to their predictive performance. The rows and columns of the panels are for the monkeys and regions, respectively. The shadowed regions correspond to percentiles 2.5 and 97.5 of the null distributions of $\tilde{R}^2_{KP}$ estimated using permutation of the stimulus labels ($N = 1000$). The horizontal arrows correspond to the median $\tilde{R}^2_{KP}$ for 2D and 3D_VD models, per monkey and region. **b** Distribution of differences between adjusted reliability-normalized coefficients of determination, denoted as

$\tilde{R}^2$, between the keypoint-based model and AlexNet for different layers of the AlexNet. *P* values from two-sided Wilcoxon signed rank tests. Columns and rows correspond to regions and monkeys, respectively. *n* indicates the number of units. Box plots show the median (horizontal line), interquartile range (box: 25th–75th percentile), and data within 1.5× the interquartile range from the lower and upper quartiles (whiskers). Points beyond this range are plotted individually as outliers. Source data are provided as a Source Data file.

captures the variation of the response of the unit explained by the model. From the PCA and beta values of the regression, we can obtain weights for each keypoint, reflecting its contribution to the selectivity of the unit (Methods). This is illustrated for an MSB unit (u258) in Fig. 3a–d, showing 25 images that produced a high (Fig. 3c) and 25 that produced a low response (Fig. 3a; Methods). When describing the poses, we will label body part-related features according to the corresponding part (arm-related features as "arm"), without implying that these units respond "semantically" to those parts. This unit responded strongly to poses with an arm lifted upwards and the tail downwards. Poses with an arm touching the (imaginary) floor caused lower responses. According to the model (Fig. 3d), the selectivity for pose was mainly driven by the position of the arm and the tail/lower back. Note that the responses to the preferred pose showed considerable tolerance for rotation around the vertical axis. The second MSB unit (u625; Fig. 3e–h) shows a different tuning, responding when the tail and lower back were oriented to the left and the head facing to the right with these parts close to a horizontal line or tilted downward. The position of the arms and legs contributed less to the pose and viewpoint selectivity. This unit was selective for rotation around the vertical axis, producing less response for the same pose when oriented to the left. An example unit from ASB (u669; Fig. 3i–l) preferred a bowing pose with the highest contribution from the arms and tail according to the 3D_VD model. This unit showed strong viewpoint tolerance.

The extent of viewpoint tolerance varied among units (Fig. 3). To quantify the viewpoint tolerance for the preferred pose, we designed a View Invariance Index (VII) that is inversely related to the range of model responses to the eight views of the preferred pose, relative to the range of the model responses to randomly chosen views of the other poses (Methods). The higher the VII, the stronger the viewpoint tolerance. The VII of the example units are indicated in Fig. 3. Since the computed preferred axis of a model is valid only for units with a good fit, we show in Fig. 4a the VII for units with a $R^2$ (unnormalized by reliability) greater than 0.25. The distributions of the VII, computed for the 2D and 3D_VD models of these units, showed a wide range in both regions of both monkeys (Fig. 4a; Supplementary Fig. S10 for all units).

The median VII of both models was significantly greater in ASB than in MSB (Fig. 4a; Supplementary Fig. S10), suggesting a greater viewpoint tolerance in the anterior region, in line with previous studies[4,6].

To provide an overview of the axes and keypoint weights of the units, we clustered the units of both monkeys using the fitted beta coefficients of the ten PCs of each unit (Methods). As for the VII, only units with an $R^2$ higher than 0.25 were included in the clustering. We will present the data for the 2D model for MSB units and the 3D_VD model for ASB units because these models provided overall the best predictive performance in the respective regions.

We obtained 15 clusters of MSB units (Supplementary Fig. S11) whereas 46 units of the 176 MSB units were not clustered (cluster "None" in Supplementary Fig. S11). Representative poses of the preferred axis for each cluster are shown in Fig. 5 (all poses in Supplementary Fig. S11). The axes of the largest cluster (cluster 4; 43 units) showed a rightward-facing profile pose of a bowing or standing monkey with the selectivity driven by the position of the tail, lower back, and to a lesser extent by the forelimbs and head. These units showed a strong viewpoint selectivity with a median VII of 0.1. The second largest MSB cluster (cluster 3; 17 units) consisted of axes mirroring those of the first cluster: a left-facing head with the tail/lower back to the right, with strongest weights for the tail, lower back, shoulder, and head, but small weighting for the limbs. As for the first cluster, its median VII was low (0.72; Fig. 5). The other clusters showed a variety of standing poses with the strongest weights for the arms, legs, and/or tail. The median VII varied among clusters, with the highest VII for vertical poses (Fig. 5).

Clustering of the ASB axes resulted in 8 clusters (Supplementary Fig. S12), with 8 of the 80 units not being clustered. The largest cluster (cluster 2; 21 units) consisted of a preferred back view of a monkey with spread and raised arms, and the selectivity was determined mainly by the position of the arm and legs. The viewpoint tolerance was moderate (median VII 2.21; Fig. 6 for a representative pose). A second cluster (cluster 4; 13 units) consisted of a sitting pose, with strong weights from the tail and lower back. The viewpoint tolerance was high with a median VII of 4.15. A third cluster (cluster 6; 13 units) had a

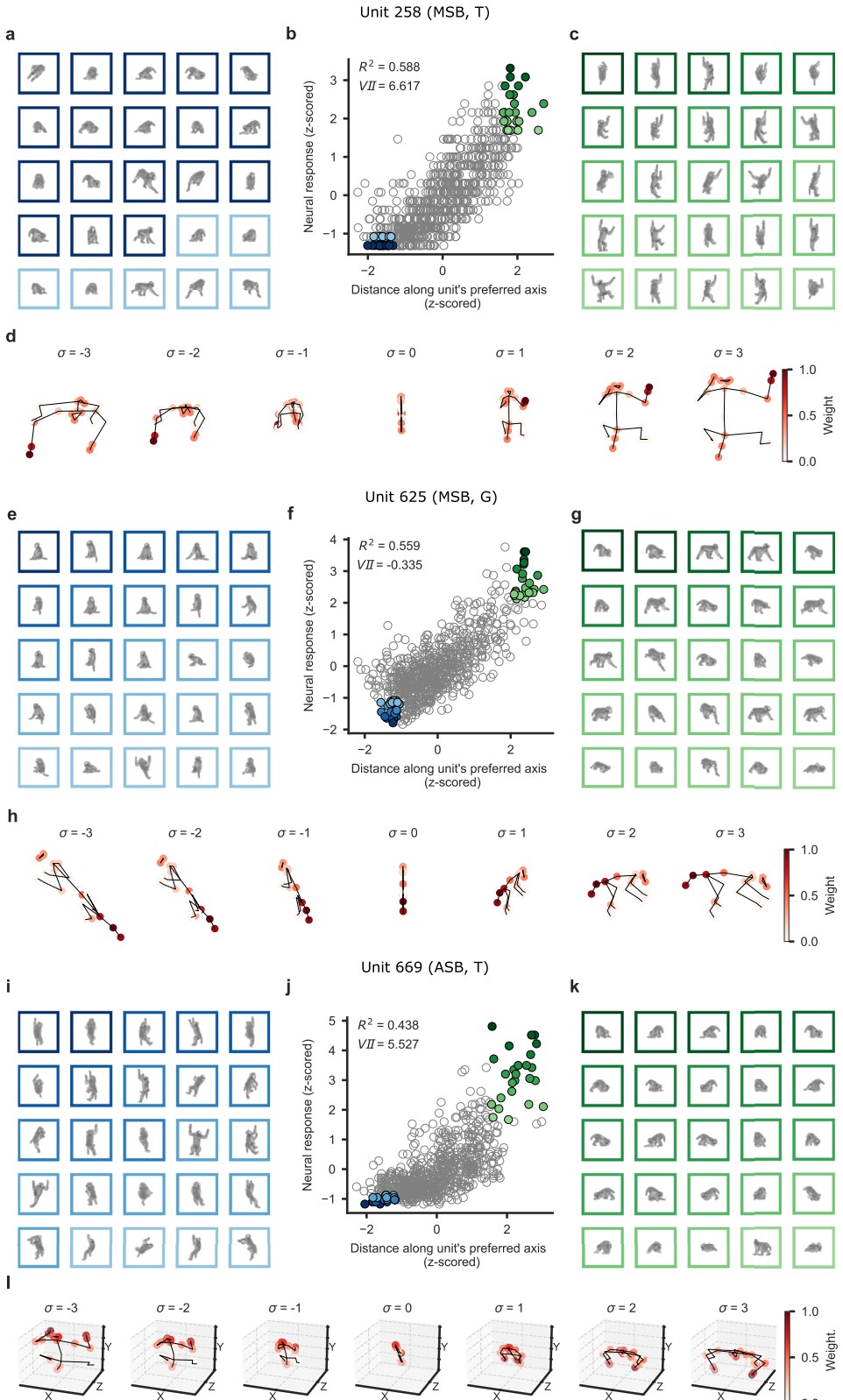

crouching pose with the head and shoulders slightly lifted and the tail down, viewed from the front (median VII −0.28). The strongest weights were for the tail/lower back, legs, head, and shoulders. The fourth largest cluster (cluster 1; 9 units) had a similar pose but with strong weights of the shoulders, arms, and hands. Also, its viewpoint tolerance was high (median VII of 5.42). Other clusters showed various standing and sitting poses (Fig. 6; Supplementary Fig. S7).

These data suggest that the pose selectivity in both MSB and ASB is driven by the relative position and orientation of body parts, like arms, legs, lower back, and tail. Which body parts are involved then depends on the unit. The axes of most units depicted a change of both orientation and pose in the PC space.

Since the 3D_VD model's z-coordinates (depth) are not explicitly represented in the 2D stimuli, we examined what makes the pose and/

**Fig. 3 | Preferred axes of units. a–d** Data for example unit 258 in the MSB region of monkey T. **b** A scatter plot depicting the relationship between the unit's distance on its preferred axis (model response) and its actual response. **c** Stimuli that elicited a high response, as shown in (**b**), are ordered in descending response strength (coded in shades of green; darker: stronger response). **a** Stimuli that elicited a low response, as shown in (**b**), are ordered in ascending response strength (blue). **d** Poses, estimated using the 2D keypoint model, corresponding to a range of standard deviations [$\sigma = -3$, $\sigma = 3$] along the preferred axis. The color of a keypoint indicates its max-normalized

weight, representing its contribution to the unit's selectivity. We randomly permuted the stimulus labels ($N = 1000$) to compute a null distribution of weights of each keypoint and computed a p-value for the observed weight. It was assigned a value of 0 when the observed weight of the keypoint was not significant (False Discovery Rate corrected). **e–h** Data for example unit 625 in the MSB region of monkey G, using the same conventions as (**a–d**). **i–l** Data for unit 669 in the ASB region of monkey T. The same conventions as in (**a–h**), except that the preferred axis was estimated using the 3D_VD keypoint model and is plotted in 3D.

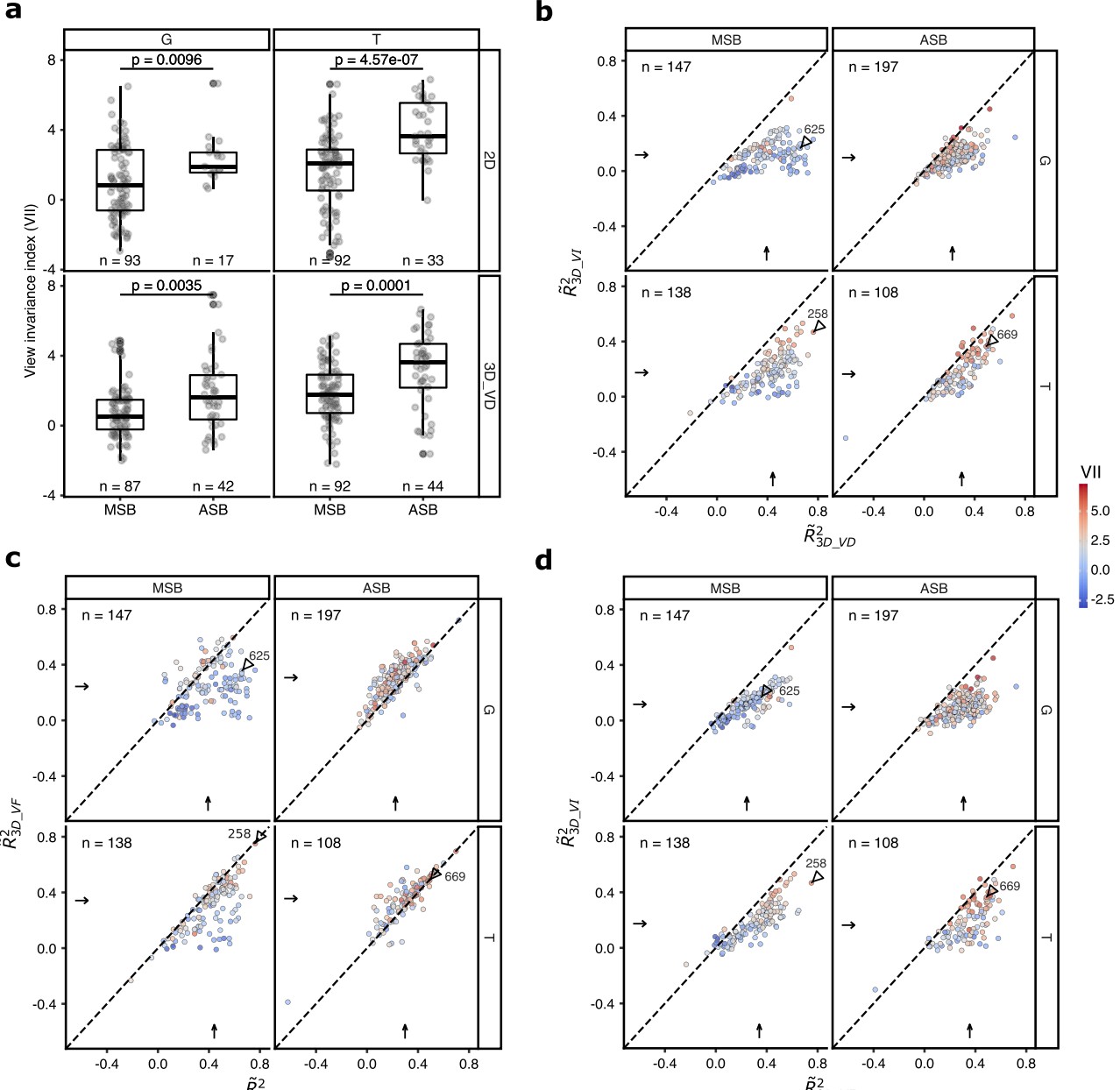

**Fig. 4 | Viewpoint tolerance. a** Distribution of the View Invariance Index (VII) of units with a coefficient of determination larger than 0.25 in the MSB and ASB regions for each monkey. The row panels display results for the keypoint models 2D and 3D_VD. *p* values are from a two-sided Wilcoxon rank sum test. **b** Scatter plot of the reliability-normalized coefficient of determination values of the view-invariant 3D keypoint model (3D_VI) versus the view-dependent 3D model (3D_VD). **c** Scatter

plot for 3D view-flipped model (3D_VF) versus 3D_VD, and **d** 3D_VI models. Each point represents a unit, with its color indicating its VII computed for the 3D_VD model. The diagonal dotted line is the identity line. Arrows indicate medians and n corresponds to the number of units. All units were included in b, c, and d. Numbers (with triangle) indicate example units (Fig. 3). Box plots follow the same convention as in Fig. 2. Source data are provided as a Source Data file.

or viewpoint selectivity special in units better fitted by the 3D_VD than the 2D keypoint model. We examined this for ASB, since the 3D_VD and 2D model predictive performances of MSB units were highly similar. First, we observed that the difference in predictive performances

between the two models ($R^2_{3D\_VD} - R^2_{2D}$) was negatively correlated with VII (Supplementary Fig. S13), indicating that the 3D_VD model of these units was highly viewpoint-dependent. Second, the preferred poses of the units with a large $R^2_{3D\_VD} - R^2_{2D}$ had an upward-slanting head and

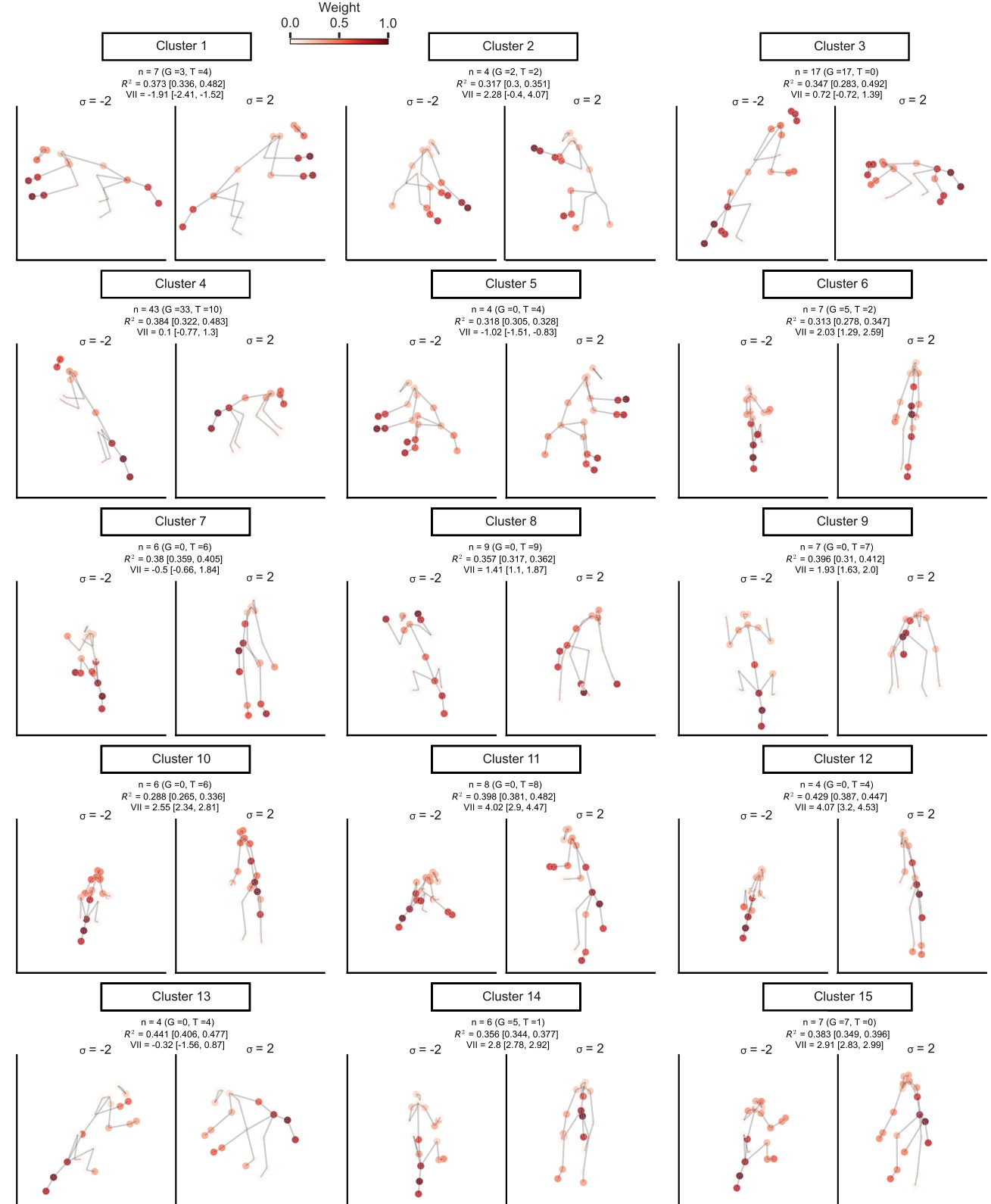

**Fig. 5 | Poses and viewpoints along the preferred axis for units in the MSB region of both monkeys.** The clusters of units are based on their preferred axes. For each cluster, a medoid representative pose is shown (see Supplementary Fig. S11 for all poses) for the standard deviation values σ = −2 and σ = 2 along the axis. Statistics of each cluster are shown, where n corresponds to the size of the cluster. The number of units is indicated by n for each cluster and each monkey (G and T). The poses were estimated using the 2D keypoint model, where the color of the keypoints represents their weight. The median coefficient of determination and VII (and percentile 25 and 75) for each cluster are shown on top of each panel.

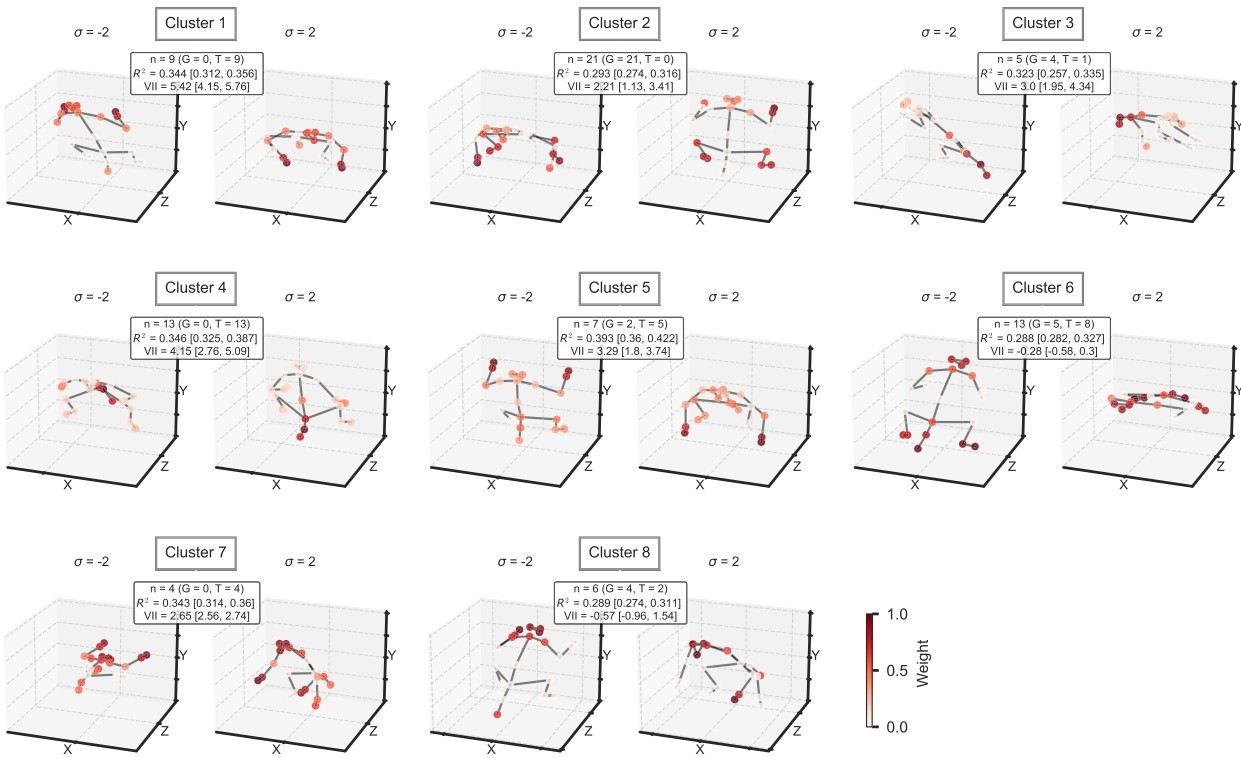

**Fig. 6 | Poses and viewpoints along the preferred axis for units in ASB of both monkeys.** For each cluster, a medoid representative preferred 3D pose is depicted (see Supplementary Fig. S7 for all poses). The poses were estimated using the 3D_VD keypoint model, with the color of the keypoints representing their weight. Same conventions as in Fig. 5.

shoulders, with a downward-slanting back and tail, and hands downward. These units were sensitive to the orientation of the body with frequent preferences for frontal or rear views (see Fig. 7 for example units; Supplementary Fig. S14). The z-coordinates of the 3D model account for such a pose variation, from rear to frontal views or vice versa. Note that these body-selective units were not sensitive merely to frontal or rear views of the head, like face cells, since the location of other body parts, particularly the tail/lower back, and legs, also contributed to their selectivity (Fig. 7; Supplementary Fig. S14).

Face-selective neurons of anterior face patch AL exhibit mirror-symmetric viewpoint tuning, conceptualized as an intermediate step toward full view invariance[23]. Our stimuli lacked true mirror-symmetrical images due to natural poses with self-occlusion and asymmetric limb positions, but profile and oblique views showed rough mirror-symmetry. To assess this tuning, we designed a 3D_VF ("view-flipped") model in which mirror-symmetric views (e.g. 90° and 270°) shared the same keypoint coordinates (Methods). Cross-validated regression with 10 PCs (Methods) showed significant predictive performances in both regions (Supplementary Fig. S15d). For MSB, the median $\tilde{R}^2$ (Supplementary Table S1) was significantly smaller for the 3D_VF than for the 3D_VD model (Fig. 4c; Wilcoxon signed rank test; G: $p = 8.13e{-}14$; T: $p = 1.45e{-}19$). However, for ASB, the median $\tilde{R}^2$ was significantly larger for the 3D_VF model (Fig. 4c; G: $p = 2.89e{-}23$; T: $p = 2.38e{-}08$; variance partitioning analysis of 3D_VF and CNN fits in Supplementary Figs. S9, S16). The improved fit for ASB was also observed for a 2D_VF model compared to the 2D model (Supplementary Table S1). These analyses suggest mirror-symmetric viewpoint tuning in ASB. ASB units better fitted by 3D_VF differed from those showing large 3D_VD vs. 2D differences (Supplementary Fig. S17), preferring axes transitioning between quasi-frontal and profile-like views (Fig. 8; Supplementary Fig. S18).

Next, we assessed the performance of a view-independent 3D keypoint model (3D_VI) by equating the 3D keypoint coordinates of the 8 views of the same pose to those of the frontal view (Methods). This model can provide only a good predictive performance when the responses are invariant to 3D pose rotation. Although the 3D_VI model performed, for the majority of units, significantly better than when permuting the stimulus labels (Supplementary Fig. S15c), it produced on average a worse fit to the data of both monkeys than the 2D, 3D_VD, and 3D_VF models (Fig. 4; Supplementary Table S1; Wilcoxon signed rank tests: maximum $p = 2.77e{-}12$ across regions, monkeys and model comparisons). The mean 3D_VI fit was similar in MSB and ASB (Wilcoxon rank sum test; G: $p = 0.05$; T: $p = 0.86$). As expected, the degree of viewpoint tolerance, as quantified by VII, correlated positively with the 3D_VI fit (Supplementary Fig. S19b) and negatively with the difference between the 3D_VD and 3D_VI models (Fig. 4b; Supplementary Fig. S19c). In sum, models that incorporated viewpoint information outperformed the view-invariant model, suggesting that few units in MSB and ASB demonstrated fully view-invariant responses to different poses.

Despite the view-dependency of the responses of many STS neurons, it is possible that different poses can be separated linearly in the population activity space of MSB and/or ASB units. If that holds, a subsequent processing stage that properly combines the activity of the MSB and/or ASB units would be able to compute poses, irrespective of viewpoint. To examine this, we used a linear classifier (Methods) to decode the poses, irrespective of viewpoint, from our sample of MSB and ASB units, after combining the units of both monkeys. Mean pose classification accuracy was 53.1% for MSB and 71.7% for ASB (standard deviations across resamples: 0.6% and 0.5%), both significantly above chance (2.22%; permutation test: Maximum mean classification score of 100 pose label permutations: 2.24% (MSB) and 2.23% (ASB)). The confusion matrices (Fig. 9) show that all 45 poses could be

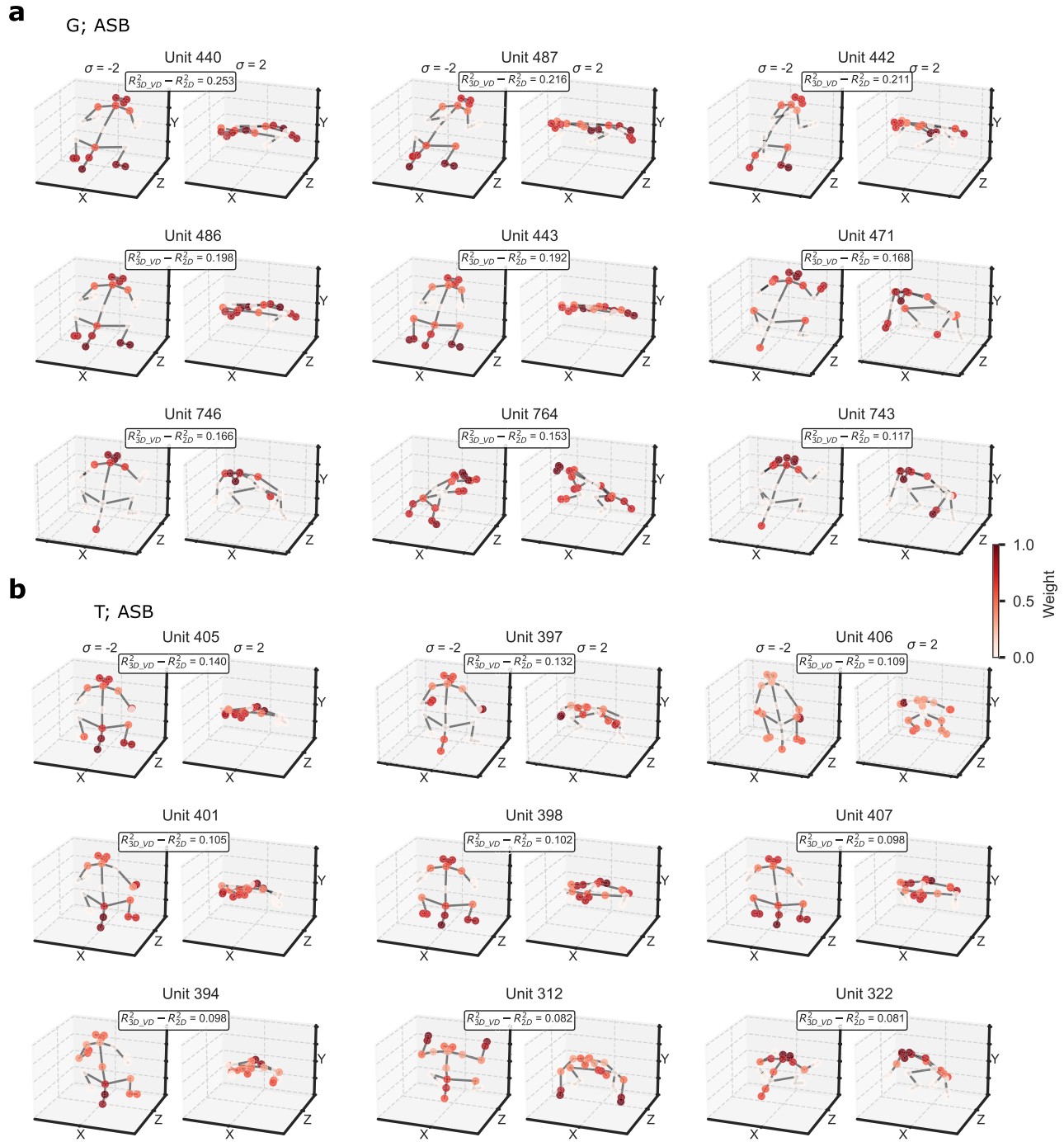

**Fig. 7 | Poses and viewpoins along the preferred axis of the top 9 units in ASB with an improved predictive performance for the 3D_VD model compared to the 2D model. a** Units of the ASB region of monkey G, with poses for standard deviation values σ = −2 and σ = 2 along the axis, where the 3D_VD model provided a better fit compared to the 2D model. The difference in R² between the 3D_VD and 2D models is shown with keypoint weights indicated by color. **b** Plots for the ASB units of monkey T. For the other units, see Supplementary Fig. S9. For each unit, R² for the 3D_VD model was larger than 0.25.

decoded, with ASB showing a higher performance and less confusion between related poses (e.g. different sitting poses).

## Discussion

We found that models using a keypoint parameterization of body pose and view explained a large amount of the response variance for monkey body poses and viewpoints of body-selective STS neurons. A 2D keypoint representation was sufficient to model the pose and viewpoint selectivity of most mid STS units. The selectivity for pose and viewpoint of different units was driven by different combinations of body parts. A

view-dependent 3D model and one incorporating mirror-symmetric viewpoint tuning provided, on average, a better predictive performance than the 2D model for anterior STS units. However, a view-independent pose model showed on average a poor fit, which is in line with the response of the majority of STS units being well described by a preferred axis that changes both in pose and viewpoint, although some units showed tolerance to the rotation of the body along its vertical axis, even in the mid STS. Despite the viewpoint dependency of the pose tuning at the unit level, the poses could be decoded, irrespective of viewpoint, when considering the response of the population of units.

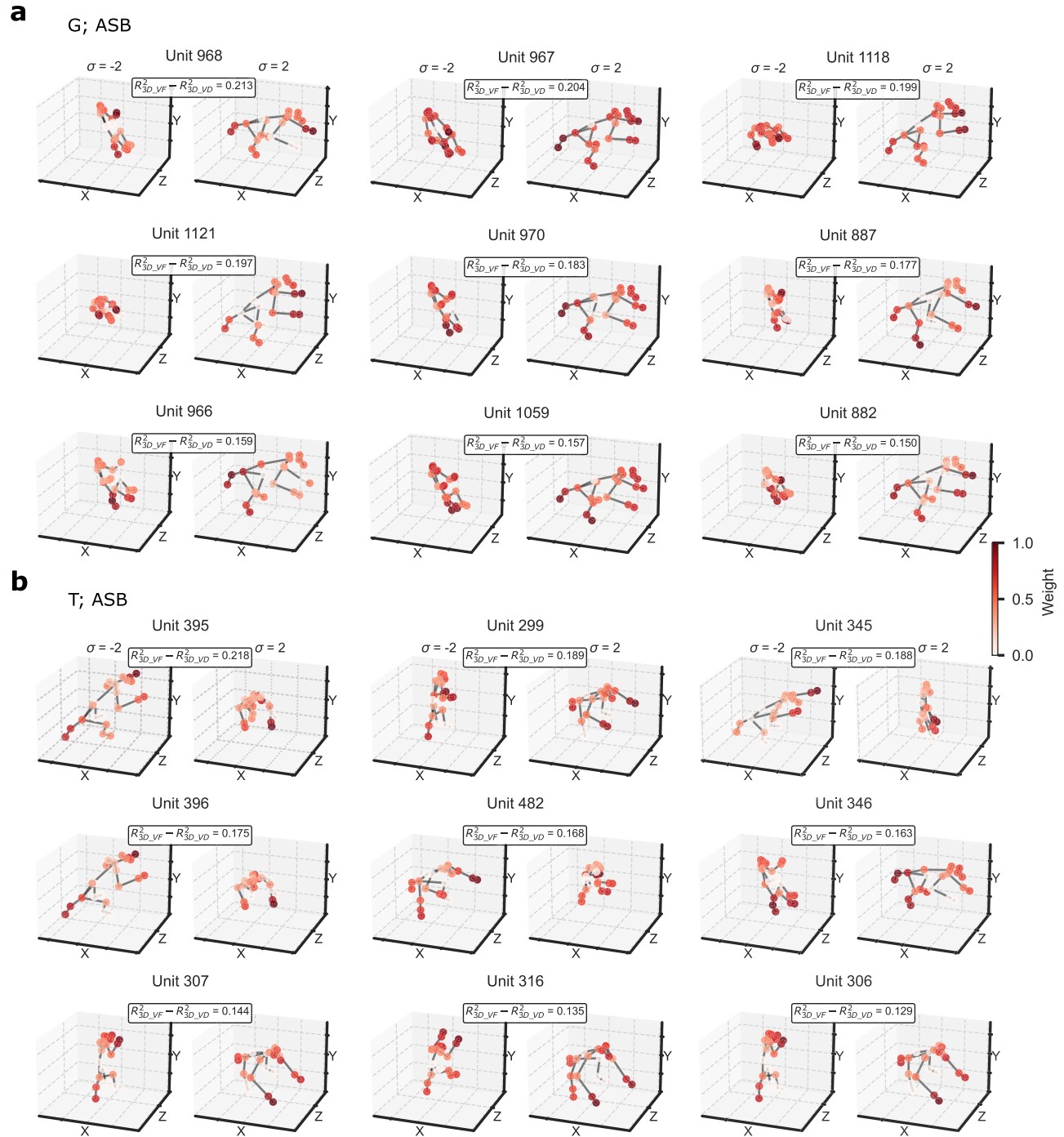

**Fig. 8 | Poses and viewpoints along the preferred axis of the top 9 units in ASB with an improved predictive performance for the 3D_VF model compared to the 3D_VD model. a** Units of the ASB region of G, with poses with σ = −2 and σ = 2 along the axis. The difference in $R^2$ between the 3D_VF and 3D_VD models is indicated for each unit, with keypoint weights shown in color. **b** Plots for the ASB units of T. For each unit, $R^2$ for the 3D_VF model was larger than 0.25.

Previous single-unit studies reported that ASB shows a greater viewpoint tolerance than MSB when considering the population of units[4,6], which aligns with the higher mean VII of ASB observed here. However, this difference in the mean viewpoint tolerance hides the variation in viewpoint tolerance that exists at the unit level in each region. Even in MSB, we observed neurons with a high tolerance for rotation of the preferred pose around the vertical axis. However, these neurons lacked complete viewpoint invariance, since they were highly sensitive to in-plane rotation of the body.

The improved average predictive performance for the models incorporating mirror-symmetric viewpoint tuning in ASB, but not in MSB, aligns with similar tuning for face viewpoint in the anterior face patch AL, but not in the middle STS face patch ML[23]. It supports computational work suggesting mirror-symmetric viewpoint tuning as a step towards view-invariance for bilateral-symmetrical objects in general[24]. The anterior IT body patch AVB[14] has been reported to show a higher degree of viewpoint tolerance than ASB[6], but this difference was smaller than between MSB and ASB, and its generalization was still poor between frontal and profile views (Fig. 3a of ref. 6). Overall, many IT neurons show view-dependent responses, with a tendency for increased view-tolerance and mirror-symmetric tuning at hierarchically later stages.

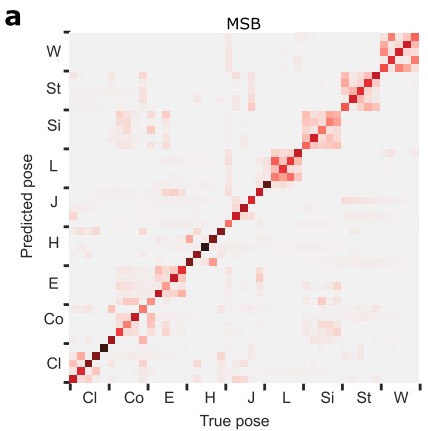
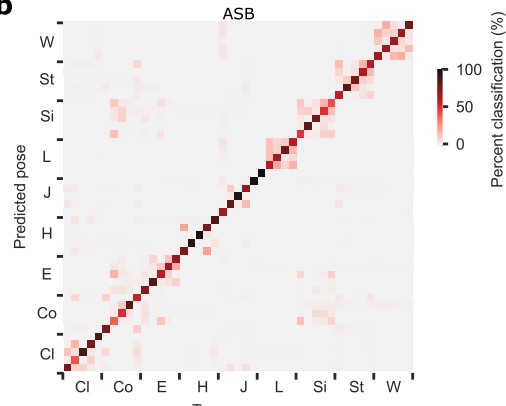

**Fig. 9 | Confusion matrices of view-invariant pose classification.** Percent classification of the pose (predicted) when the trained linear classifier was fed with the responses of a single presentation of a pose (true). Percent classification sum to 100 for each column. **a** MSB; **b** ASB. Units of both monkeys were pooled. Poses are indicated by their class (Fig. 1a). Source data are provided as a Source Data file.

The preferred axis of many units corresponded to changes in both pose and viewpoint. This might not be surprising given that the combination of pose and body orientation has strong ecological significance, e.g., body orientation can signal the target of a gesture, and one would expect this to be encoded even at higher processing stages. This might explain why even in the anterior STS we observed units with responses strongly dependent on whether the avatar was facing the observer.

The view-independent model fitted worse than the view-dependent models, which is in line with the contribution of both viewpoint and pose to the axes of the units. Nonetheless, a linear decoder, trained on the responses of the population of recorded MSB or ASB units, was able to classify the poses well above chance, irrespective of viewpoint. This suggests that pose can be read out in a view-invariant fashion, from the population of units, especially from ASB. The viewpoint-dependency of the responses enables also a readout of viewpoint and particular pose-viewpoint combinations from the same population of units at different hierarchical stages. Thus, in principle, the output of the same population of units can be weighted differently by different higher-order areas depending on whether they need to process body orientation, pose, or both.

A human fMRI study[25] suggested a view-independent pose representation in the anterior part of EBA, a potential homolog of macaque MSB[27]. However, they employed only three viewpoints, with a maximal difference of 90°. Furthermore, most of the combined models' variance was explained by the 2D keypoint model, consistent with our MSB data. Another human fMRI study[26], employing Representation Similarity Analysis[28] (RSA), found a correlation between their 3D_VI pose model and the neural representations of about 0.02 in EBA and lower in the FBA, a potential homolog of ASB[14]. Correlations for view-dependent pose models tended to be somewhat larger. We find these low RSA correlations difficult to interpret.

Unlike fMRI studies, our model-based approach revealed the tuning for pose and viewpoint of units. The preferred axes could be clustered in different groups varying between regions. For instance, few units of our MSB sample preferred sitting or eating poses, while many units preferred poses with extended limbs. It remains unclear whether this reflects a genuine difference in pose preference between regions or a sampling bias, which can be larger when using linear probes due to columnar clustering of stimulus preferences in IT[29–31], which was also evident in our study (Supplementary Fig. S20).

The 2D and 3D_VD predictive performances were greater in MSB than in ASB. The keypoint representation is a rather abstract representation of the spatial configuration of body joints and body part features, ignoring the texture, shading, and local form of those.

ASB may be more sensitive to texture and/or local form than MSB (Supplementary Fig. S9), but this requires experimental verification. Interestingly, the best fit of the CNNs across layers was also greater for MSB than ASB units, suggesting that the difference in model fits between the two regions is not specific to the keypoint representation. Note that the differences between regions cannot be explained by reliability of the selectivity, since both regions had highly similar reliability.

Our keypoint-based models have as input labeled keypoint coordinates which are not readily available in the image but require computation, involving (nonlinear) operations to detect body parts and/or joints. Our modeling shows similarities to the "axis model" of face-coding in face patches[22], which is based on PCA dimensions of keypoints of inner face features and shape-normalized luminance ('appearance') features, which are also not directly available from an image. However, recent computer vision algorithms are able to compute keypoints from a body image[32], showing that, in principle, keypoint-based models are image-computable. Some ASB units were fitted better by the view-dependent 3D model than by the 2D model, although the z- (depth) coordinates of the keypoints in our 2D images require inference. Further research is needed to examine whether this inference is based on pictorial depth cues[33,34], the statistical structure[35] of body configurations in the 2D images, and/or knowledge of 3D body configuration[36].

We used PCA to reduce the dimensionality of the keypoint variations of the stimuli and to obtain uncorrelated dimensions for regression, without implying that the PCs are the main dimensions of the neuron's response space. The advantage of our keypoint-based models is that they allow a quantification of the selectivity for pose and viewpoint as an axis in a relatively low-dimensional PC space. Moreover, by inverting the models, one can visualize the pose and viewpoint changes that drive the selectivity of the unit for a large stimulus set. This allowed us to identify the body parts that drive the selectivity. Our work demonstrates the power of modeling the responses of units to large stimulus sets to reveal aspects of the neural tuning for pose and viewpoint, which would be difficult to assess by examining responses to a wide variation of body stimuli without any model.

## Methods
### Subjects
Two male rhesus monkeys, G (7 years old) and T (7 years old), served as subjects. The monkeys were implanted with an MRI-compatible plastic headpost and a custom-made MRI-compatible chamber (G: right hemisphere; T: left hemisphere). The implantations were performed under general propofol anesthesia and aseptic conditions. The

recording chamber was positioned to have access to all STS body patches. Animal care and experimental procedures complied with the regional (Flanders) and European guidelines and were approved by the Animal Ethical Committee of KU Leuven.

## Recording locations

We employed fMRI for targeting the recording locations in each monkey. We employed a block design with six conditions: 20 dynamic monkey bodies, 20 dynamic monkey faces, and 20 dynamic objects, and mosaic-scrambled controls of each. The movies, block design, and experimental procedures were identical to that of ref. 12. The body, face, and object stimuli were presented on top of a dynamic white noise background. The monkeys were scanned with a 3T Siemens Trio scanner while fixating a small fixation target (fixation window 2°–3°). The movies were projected on a translucent screen (refresh rate 60 Hz; for details see[12]) positioned 58 cm in front of the monkey. Scanning was performed following intravenous injection of the contrast agent Monocrystalline Iron Oxide Nanoparticle (MION). Each block consisted of twenty 1 s long movies. Each block was presented twice in a run, interleaved with 3 baseline blocks in which only the fixation target and white noise background were presented, using a palindromic design.

The data were analyzed in each monkey's native space, obtained from a high-resolution (400 um) structural MRI scan of each monkey which was obtained before the fMRI scanning. Standard preprocessing steps and analyses (General Linear Model using SPM12) were employed to compute t scores of the contrast bodies minus faces and objects. For details of the standard analysis steps, see[12]. Voxels with $t > 4.96$ (Family-Wise corrected $p < 0.05$) were considered significant. Activation maps were visualized using FSLeyes for each monkey's brain.

We defined body patches with the contrast bodies minus faces and objects. In each monkey, we observed a body patch in the medial part of the ventral bank of the mid STS (MSB), and a second body patch more anterior in the ventral STS (ASB). In T, MSB activations in the recorded left hemisphere were weaker than in its right hemisphere, but subsequent multi-unit recordings showed a body-selective cluster of sites close to the left hemisphere MSB activations in this monkey. T's ASB of its left hemisphere was more posterior than G's. Given the difference in ASB location between monkeys, we present the recording data of each monkey separately in most analyses.

## Recordings and procedures

Recordings of spiking activity were performed using 16 channel V probes (185 μm diameter, 100 μm electrode spacing; Plexon). The probe was lowered through a stainless-steel guide tube that was positioned in a custom-made plastic grid and only after at least one hour of waiting the data were obtained to ensure recording stability. We targeted body-selective clusters of neurons in or close to the MSB and ASB patches defined in the fMRI. We will label the units recorded from these clusters as MSB and ASB units, respectively. The co-registration of the anatomical images with fMRI activations and recording site targeting were described in[13,16]. The analyzed recording data are from 11 and 12 penetrations for the MSB region of monkey G and T, respectively, and 15 and 6 penetrations for ASB of G and T, respectively.

The recorded broadband signals were amplified using an INTAN headstage, band-passed filtered (500 Hz–5 kHz), and displayed online with an Open Ephys recording system. The filtered signals, together with the timings of the stimulus events, were saved at a 30 kHz sampling rate. Single units and multi-unit clusters were sorted with Plexon Offline Sorter 4.6.2. using manual PCA-based sorting of thresholded waveforms. We will use the term "unit" to refer to both single units and multi-unit clusters, i.e. "unit" does not imply a single unit.

During the recordings, the monkey was seated head-fixed in a primate chair facing a VIEWPixx LCD display from a distance of 58 cm. Eye position was recorded with a video-based eye tracker (EyeLink 1000). The stimulus on- and offsets were measured with a photodiode detecting the transition of a small square from black to white (invisible to the monkey), coinciding with stimulus events, in the upper left corner of the display. A custom-made system controlled the stimulus presentation and behavioral paradigm, delivered juice rewards, and saved the stimulus and reward events and eye movement traces (sampled at 2 kHz) (see[4] for further details).

## Stimuli and tests

**Body-category selectivity.** To assess body category selectivity, we employed 20 movies of acting monkeys, 20 movies of dynamic monkey faces, and 20 movies of dynamic artificial objects. Each movie was 1 s long. The movies were identical in the electrophysiology and fMRI body localizer experiments and matched those described by Bognar et al.[12], except that in the electrophysiology experiments the bodies, faces and objects were presented on an empty gray background instead of on a white noise background in some of the tests of G and all tests of T. The heads of the acting monkeys were blurred. The movies had a frame rate of 60 Hz and were presented with a refresh rate of 120 Hz on a VIEWPixx LCD monitor.

The 60 movies were presented in random order during fixation of a small square fixation target (0.2°) in blocks of 60 unaborted presentations. Each movie was presented once per block, requiring fixation of 200 ms before, during, and 200 ms after the presentation, otherwise saved as an aborted trial. Aborted movies were presented again at a random time during the same block. Five unaborted presentations of each of the movies were collected. Monkeys received an apple juice reward using a fixed interval reinforcement schedule when maintaining fixation, titrated per monkey. The fixation window was approximately 2° on a side.

**Pose and viewpoint selectivity.** The main stimulus set consisted of 720 images of a monkey avatar displaying 16 views of 45 natural poses. The 45 poses can be grouped into 9 classes of behavior, like sitting, eating, hanging, laying, standing, and walking (Fig. 1). However, we treat each of the 45 poses as separate since the classes are based on semantic labels that may differ between monkeys and humans. The 16 views consisted of 8 orientations (45° steps of azimuth) of a horizontally positioned camera and 8 orientations (45° steps of azimuth) from a 45° tilted camera elevation angle. The set of natural monkey poses was obtained using the OpenMonkeyStudio deep learning-based markerless motion capture system[5]. These were then processed using Autodesk MotionBuilder and Maya. The motion capture data from OpenMonkeyStudio provided us with a sparse set of initial 3D keypoints that were used as a reference in the 3D modeling software Maya to position the skeleton of our custom-made monkey avatar. The corrections of skeletal poses, meshes (blend shape mixtures), and fur were performed manually for each of the poses using Maya. The images were rendered in Maya using the Arnold renderer. The camera axes were aimed at the pelvis center of the monkey avatar. For the 8 horizontal camera positions, this point is visible horizontally at the same distance for all cameras. The corresponding 8 tilted camera positions have the same aim and distance to the aim point, but they were elevated by 45° (i.e., they were looking down 45°), except for the "hanging up on the ceiling" poses where those 8 camera angles were displaced down, below the horizontal plane, looking up with a tilt of 45°. The coordinates of the keypoints were computed orthographically aligned with the direction of the camera. The color images underwent color saturation, gamma, and contrast corrections.

The images shown to the monkeys were achromatic, gray-level versions of the original color images and were scaled so that the maximum horizontal and vertical extent across all 720 images was 6°. They were presented gamma-corrected on a VIEWPixx display for 200 ms preceded by a 250 ms long fixation interval during which only the fixation target and the gray background were presented. The

monkeys were performing a fixation task for a juice reward using a fixed interval reinforcement schedule. The 720 images were presented in blocks of 720 unaborted presentations. In total, 6 and 8 unaborted presentations of each stimulus were collected for each unit in monkeys T and G, respectively. Only presentations in which the monkey fixated during the pre-fixation period of 250 ms and during the stimulus were analyzed, except for 1.3% of the presentations in T during which the monkey was fixating for 150–200 ms during the stimulus presentation.

### Data analysis

**Selection of units.** We employed the following criteria to select units that were selectively responsive to pose and/or viewpoint. (a) We compared the firing rates between the baseline (window of 100 ms, starting 75 ms before stimulus onset) and responses (response window of 200 ms, shifted by 50 ms to account for response latency) using a Split-Plot ANOVA with the two windows as a repeated factor and the stimulus condition as a between-trial factor. A unit was kept when either the main effect of the window and/or the interaction of the two factors was significant ($p < 0.05$). These analysis windows were the same in all subsequent analyses. (b) To assess signal stability, we computed the Fano factor of the spike count during stimulus presentation using linear regression of the trial-to-trial variance in spike count versus the mean response per stimulus. Units were required to have a Fano factor smaller than 4. (c) The mean net firing rate to a stimulus had to exceed 5 spikes/s. (d) Selectivity for pose and/or viewpoint was assessed with a Kruskal–Wallis ANOVA on the firing rates in the response window. Only units with a significant effect of stimulus condition (pose or viewpoint) were kept ($p < 0.05$). (e) We computed the reliability of each unit as the Spearman-Brown corrected split-half Pearson correlation coefficient (the number of trials split in two per stimulus). Only units with a reliability greater than 0.5 were analyzed further. These criteria were applied simultaneously using an "AND" operation. The implementation of these criteria ensured proper modeling of pose/viewpoint selectivity for units with reliably different responses across poses/viewpoints. Our selection criteria for responsivity, selectivity, and reliability retained 53% of sorted units on average, across monkeys, for both MSB and ASB —coincidentally the same percentage for each region. This yielded 346 reliably selective MSB units (average per penetration: 10.5 units) and 544 ASB units (average: 25.9 units).

For each electrode site, we computed the Body Selectivity Index based on the mean net firing rates for the bodies, faces, and objects obtained in the body-category selectivity test:

$$\text{BSI} = (\text{mean net firing rate bodies} - \text{mean net firing rate non-bodies})/$$
$$(|\text{mean net firing rate bodies}| + |\text{mean net firing rate on-bodies}|). \quad (1)$$

The net firing rate was computed using a baseline window of 200 ms and a response window of 1 s, delayed by 50 ms to account for the response latency. Only units from an electrode site with a BSI equal or larger than 0.33 (twofold stronger response to bodies compared to faces and objects[13,37]) that showed a significant response to the stimuli (Split-Plot ANOVA; same design as above; $p < 0.05$) were modeled. This applied to 76% of reliable, selective MSB units in G and 91% in T, as well as 49% of ASB units in G and 78% in T. Thus, for MSB, this resulted in 285 modeled units (average yield of 8.6 units per penetration), and for ASB, in 305 units (14.5 per penetration). We employed the body-category selectivity criterion to exclude units from face-category selective sites and sites for which the average object and face versus body responses were similar.

**Decoding analysis.** We decoded the 45 poses irrespective of the viewpoint, using six presentations per viewpoint-pose combination, which allowed us to pool the data of the two monkeys for MSB and ASB.

The decoder was fed the mean firing rate of single stimulus presentations, averaged in a window of 200 ms that started 50 ms after stimulus onset. The presentations of the 16 views of a pose were assigned the same stimulus label. Pseudo-population responses for a pose were created by random concatenation of the unit responses for single presentations of that pose, i.e. a vector with the length number of units of a region. The classifier was trained to decode the stimulus label using a leave-one-out cross-validation procedure. We employed the max-correlation classifier as implemented in the Neural Decoding toolbox[38]. We used 50 resamplings. We report the mean classification scores of the independent test data across cross-validations and resamplings. To assess statistical significance, we permuted the stimulus labels and reran the classification procedure 100 times. We report the maximum mean classification score of the 100 permutations. Additionally, we ran a linear Support Vector Machine classifier, which produced highly similar classification scores (means: MSB: 54%; ASB: 69%).

### Models

**Keypoint data matrix.** For the 2D model, we reshaped the keypoint data of 22 keypoints of the 720 stimuli into a keypoint data matrix $X \mathbb{R}^{720 \times 44}$, by concatenating the $x$ and $y$ coordinates of each keypoint along the columns and stimuli in rows:

$$X = \begin{bmatrix} x_1^1 & y_1^1 & x_2^1 & y_2^1 & \cdots & y_{22}^1 \\ x_1^2 & y_1^2 & x_2^2 & y_2^2 & \cdots & y_{22}^2 \\ \vdots & \vdots & \vdots & \vdots & \ddots & \vdots \\ x_1^{720} & y_1^{720} & x_2^{720} & y_2^{720} & \cdots & y_{22}^{720} \end{bmatrix} \in \mathbb{R}^{720 \times 44} \quad (2)$$

Similarly, for the 3D_VD model, we reshaped the data into a data matrix $X \mathbb{R}^{720 \times 66}$ by concatenating $x$, $y$, and $z$ coordinates along the columns. We divided each coordinate value by the image size of 350, scaling all coordinate values to fall within the range of 0 to 1.

We created a view-flipped keypoint data matrix for the view-flipped (VF) models. For each pose and camera elevation, we replaced the rows of the 2D and 3D_VD matrices —corresponding to 2D_VF and 3D_VF models, respectively— for the views at 315°, 270°, and 225° with those at 45°, 90°, and 135°, respectively. The coordinates for the views at 0° and 180° remained unchanged, giving

$$x^{view=315°} = x^{view=45°}; x^{view=270°} = x^{view=90°}; x^{view=225°}$$
$$= x^{view=135°} \in \begin{cases} \mathbb{R}^{1 \times 44}(2D) \\ \mathbb{R}^{1 \times 66}(3D) \end{cases} \quad (3)$$

As a control, we reversed this process such that the views at 45°, 90°, and 135° were replaced with those at 315°, 270°, and 225°, respectively. The latter yielded highly similar results (the median $R^2$ for the 3D_VF model was equal up to the second decimal place for the two approaches) and we will report the data using the first approach.

To create a view-invariant keypoint data matrix $X$ for the 3D_VI model, we started with the 3D_VD matrix, which contains keypoint data for various viewpoints. We selected the keypoint data corresponding to the 0° view for each camera elevation angle as the reference. We then repeated these rows for the remaining views (45°, 90°, …, 315°) for each pose and camera elevation. Consequently, each row in the resulting matrix $X$ contains identical keypoint data, making

$$x^{view=45°} = x^{view=90°} = \ldots = x^{view=315°} = x^{view=0°} \in \mathbb{R}^{1 \times 66} \quad (4)$$

**CNN data matrix.** We employed two CNNs, AlexNet[19] and ResNet50_rbst[20]. The weights for AlexNet (AlexNet_Weights.IMAGENET1K_V1) were imported from TorchVision in PyTorch, which had been pre-trained with ImageNet data. ResNet50_rbst was adversarially trained on ImageNet and has demonstrated good alignment with neural data[39].

To obtain the activations of the CNN units, we presented each of the 720 stimuli to the networks. The stimuli were pre-processed by rescaling and normalizing using the mean and standard deviation of the ImageNet data (the same pre-processing as during training). For the AlexNet models, we obtained activations from all seven (5 convolutional and 2 fully connected) layers. For the ResNet50 architectures, we collected activations from the convolutional layers at the end of each of the five stages (conv1, layer2.3.conv3, layer3.5.conv3, layer4.2.conv3), which are labeled as layers 1–5 in the Results section. For each layer, we created a data matrix $X \mathbb{R}^{720 \times M}$ from the activations of M units in that CNN's layer to the 720 stimuli.

**Pixel-based model.** We first downsampled a silhouette version of the 720 pose stimuli from $350 \times 350$ pixels to $25 \times 25$ pixels such that it retains the overall shape information in a lower dimension. Then we created a data matrix $X \mathbb{R}^{720 \times 625}$ by vectorizing the pixel values and arranging them into the columns of the data matrix.

## Modeling

We first reduced the dimensionality of the data matrices by performing PCA and then used the principal components as predictors and the unit's response as the dependent variable to fit a regression model for the unit.

We centered each of the columns of the data matrix by demeaning it and then applied the PCA to it. For the main analyses, we selected the first 10 principal components, $P$, for the keypoint-based model. For CNNs and the Pixel-based model, we selected the first 50 principal components. We also tested model fits for different numbers of selected PCs for the keypoint 2D and 3D_VD (up to 30 PCs; Supplementary Fig. S5) and CNN models (ranging from 10 to 200; Supplementary Figs. S6, S7). For the view-flipped (3D_VF) and view-independent (3D_VI) models, we employed also PCA to reduce the dimensionality to 10 (cumulative explained variance 3D_VF model: 87%; 3D_VI model: 90%). Employing a complementary approach in which we kept the original 3D_VD PC space, but equated the PC scores of the mirroring views for the 3D_VF models, followed by regression, produced highly similar predictive performances for the 3D_FV model in MSB and ASB, compared to the first approach in which the PCA was performed on the view-flipped keypoint coordinates (Pearson correlations between both approaches: $r > 0.998$). The same was true for a 3D_VI model when equating PC scores for the views instead of their keypoint coordinates before the PCA (first approach). This shows that the differences in predictive performances between the 2D/3D_VD and the 3D_VF and 3D_VI models do not result from PCA space differences. We report data from the first approach because this allows visualization of the preferred axes of the units.

**Model fitting.** For each unit, we performed 10-fold cross-validated regression on the z-scored response $y_k$ to the stimuli $S_k$ of the $k^{th}$ fold. The response to a stimulus was defined as the across-trial averaged firing rate within a window of 200 ms that started 50 ms after stimulus onset. The regression coefficients $\hat{\beta}_k$ for the $k^{th}$ fold, can be obtained as:

$$\hat{\beta}_k = (Z_k^T Z_k)^{-1} Z_k^T y_k \tag{5}$$

where $Z_k = X_k P$ corresponds to the principal component scores of the 648-model data for the $k^{th}$ fold. The model prediction for the 'out of fold' model data $X_{k'}$ is:

$$\hat{y}_{k'} = X_{k'} P \hat{\beta}_k \tag{6}$$

We then computed the coefficient of determination as:

$$R_k^2 = 1 - \frac{\sum (y_{k'} - \hat{y}_{k'})^2}{\sum (y_{k'} - \bar{y}_{k'})^2} \tag{7}$$

where, $y_{k'}$ is the unit's response to out-of-fold stimuli $S_{k'}$. And $\bar{y}_{k'}$ denotes the mean of $y_{k'}$. We obtained the final values of betas $\hat{\beta}_k$, which represents the preferred axis of the unit, and $R^2$, the coefficient of determination, as an average value of $\hat{\beta}_k$ and $R_k^2$ across the folds.

For inter-model comparisons, we computed the adjusted $R_{adj}^2$

$$R_{adj}^2 = 1 - \frac{(1-R^2)(N-1)}{N-M-1} \tag{8}$$

where $N = 720$ stimuli and $M =$ number of predictors in the model. We corrected the coefficient of determination values for the units' reliability by dividing the $R^2$ and $R_{adj}^2$ by the reliability of the unit, computed as the Spearman–Brown corrected split-half Pearson correlation coefficient. The reliability-normalized values are denoted as $\widetilde{R}^2$ and $\widetilde{R}$ for the non-adjusted and adjusted coefficients, respectively. We report also unadjusted $\widetilde{R}^2$ for the models with a different number of predictors (Supplementary Figs. S5, S7).

Null distributions of $\widetilde{R}^2$ were obtained by permuting the stimulus labels (1000 times) before performing the multiple regression. We employed also two other approaches. In the first alternative approach, we shuffled the stimulus labels only for the test stimuli after fitting the model using the original labels (Supplementary Fig. S15a). With this procedure, the $\widetilde{R}^2$ values for the shuffled test data tended to be even lower than when shuffling the stimulus labels before fitting the model. In the second alternative approach, we permuted the beta weights across the 10 PCs when predicting the responses after fitting the model (Supplementary Fig. S15b). The coefficient of determination will be negative when the model fits the data worse than just taking the mean of the dependent variable. This can happen after permutation because the original relationship between the predictors and the dependent variable is destroyed.

**Variance partitioning.** To perform the variance partitioning analysis, we fitted a combined model by concatenating the PC scores **Z** from both a keypoint and a CNN model as input to a 10-fold cross-validated regression model with 60 predictors (10 from the keypoint (KP) model and 50 from the CNN). We then computed $R^2$ for this combined model. In addition, we computed the $R^2$ for each model separately, as above. The unique and shared variance contributions of the two models were calculated as follows:

$$R_{KP\,unique}^2 = R_{Combined}^2 - R_{CNN}^2 \tag{9}$$

$$R_{CNN\,unique}^2 = R_{Combined}^2 - R_{KP}^2 \tag{10}$$

$$R_{Shared}^2 = R_{KP}^2 + R_{CNN}^2 - R_{Combined}^2 \tag{11}$$

**Feature visualization.** To interpret what a principal component represents, we computed and visualized (Supplementary Figs. S2b, S3b) the "eigenposes" $e_i$ along the $i_{th}$ principal component $p_i$, for different steps $\kappa = [-3, 3]$ of standard deviation $\sigma_i$ (square root of the eigenvalue of $p_i$):

$$e_{i\kappa} = \bar{X} + \kappa \sigma_i p_i \tag{12}$$

where $\bar{X}$ is the mean vector of each keypoint coordinate across the 720 samples. $\kappa \sigma_i$ is the movement along the principal component directions in steps defined by standard deviations.

For the 2D case, $e_{i\kappa} \in \mathbb{R}^{44 \times 1}$ is reshaped to a matrix $E_{i\kappa} \mathbb{R}^{22 \times 2}$ such that the first and second columns are $x$ and $y$ coordinates of 22 keypoints. Similarly for 3D_VD, we had $E_{i\kappa} \mathbb{R}^{22 \times 3}$ such that the first, 2nd, and 3rd columns are $x$, $y$, and $z$ coordinates of 22 keypoints.

**Preferred axis.** The preferred axis $\hat{\beta}$ of a unit lies in a 10-dimensional space in which each dimension corresponds to a principal component.

We estimated the contribution of each keypoint as the sum of the principal components weighted by their corresponding $\hat{\beta}$

$$\hat{t} = P\hat{\beta} = \sum_i \hat{\beta}_i \, p_i \in \mathbb{R}^{44} \tag{13}$$

We reshaped the vector $\hat{t}$ to a matrix $T \in \mathbb{R}^{22 \times 2}$, such that the first and second columns, $t_x$ and $t_y$, correspond to the $x$ and $y$ coordinates of the 22 keypoints. The contribution, referred to as the weight, of the $i_{th}$ keypoint is defined by the magnitude of the row vector

$$w_i^{2D} = \sqrt{\left(t_x^i\right)^2 + \left(t_y^i\right)^2} \tag{14}$$

Similarly, for 3D_VD

$$w_i^{3D\_vd} = \sqrt{\left(t_x^i\right)^2 + \left(t_y^i\right)^2 + \left(t_z^i\right)^2} \tag{15}$$

For each unit, the preferred pose, at $\kappa$ standard deviation along the preferred axis $\hat{\beta}$ was estimated as:

$$a_\kappa = \bar{X} + \sum_i \hat{b}_i \kappa \sigma_i p_i \tag{16}$$

Where $\hat{b}_i = \sigma_i \hat{\beta}_i$. Essentially, $\sigma_i \hat{\beta}_i$ scales the $\hat{\beta}$ from the PCA-transformed space (where the principal components are unit vectors) to units of standard deviation. Like eigenposes, for the 2D case, $a_\kappa \in \mathbb{R}^{44 \times 1}$ was reshaped to a matrix $\in \mathbb{R}^{22 \times 2}$ such that the first and second columns are the $x$ and $y$ coordinates of the 22 keypoints. Similarly, for 3D_VD, we had a matrix $\in \mathbb{R}^{22 \times 3}$ such that the first, 2nd, and 3rd columns are the $x$, $y$, and $z$ coordinates of the 22 keypoints.

**View invariance index.** To estimate the view invariance of the model, we first computed the model unit's response to a keypoint data matrix, denoted as $\hat{y}$. These are also the stimulus distances along the preferred axis of the neuronal unit. From these responses, we identified the pose and camera elevation angle that elicited the highest response value, $\hat{y}_{max}$. We then calculated the range of response values for all 8 views of this combination, defining this as the observed range, $r_o$. We, then, randomly sampled 7 response values from the remaining 712 data points and included $\hat{y}_{max}$ in this set to calculate their range. This random sampling and range calculation were repeated 1000 times to build a distribution, $r_n \in \mathbb{R}^{1000}$. Subsequently, we standardized this null distribution by converting it into Z-scores. The mean $\mu_n$ and standard deviation $\sigma_n$ of the null distribution were computed. The Z-score of the observed range value was then calculated as:

$$Z_o = \frac{r_o - \mu_n}{\sigma_n} \tag{17}$$

Finally, the negative of this Z-score was defined as the view invariance index (VII), given by:

$$VII = -Z_o \tag{18}$$

This index quantifies the model's tolerance to changes in viewpoint.

**Clustering units according to their preferred axes.** To concisely present the preferred pose of all units, we used Density-Based Clustering with Hierarchical Density Estimates (HDBSCAN) [https://link.springer.com/chapter/10.1007/978-3-642-37456-2_14] implemented in Scikit-Learn [https://scikit-learn.org/]. This method clusters the units based on their preferred axis $\hat{\beta}$. We set min_cluster_size to 2, meaning a cluster must contain at least two samples. Additionally, we set min_samples to 2, so a point is considered a core point only if there are at least two samples in its neighborhood.

Data were analyzed using MATLAB R2017b, R2023b, and Python 3.9.7.

**Reporting summary**
Further information on research design is available in the Nature Portfolio Reporting Summary linked to this article.

## Data availability
The data from which the figures and statistics are derived are available at Code Ocean. The data is at https://codeocean.com/capsule/2215416/tree. Source data are provided with this paper.

## Code availability
The code to generate the figures and the axes for each unit is at https://codeocean.com/capsule/2215416/tree.

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

## Acknowledgements

The authors thank W. Depuydt, C. Fransen, I. Puttemans, A. Hermans, C. Ulens, S. Verstraeten, K. Lodewyckx, J. Helin and M. De Paep for technical and administrative support. We thank Alexander Lappe for feedback on the methodology and comments on an earlief draft. This research was supported by Fonds voor Wetenschappelijk Onderzoek (FWO) Vlaanderen (G0E0220N; R.V.), KU Leuven grant C14/21/111 (R.V.), the European Research Council (ERC) under the European Union's Horizon 2020 research and innovation programme (grant agreement No. 856495; R.V. & M.G.). L.M is also supported by the International Max Planck Research School for Intelligent Systems (IMPRS-IS).

## Author contributions

Conceptualization, R.V., A.B., R.R.; Methodology, R.V., A.B., R.R., A.M., L.M., M.G.; Investigation, R.V., R.R., A.B. and G.G.N.; Data analysis, R.R, A.B., R.V.; Writing – Original Draft, R.V., R.R and A.B.; Writing – Review & Editing, R.V., R.R., A.B., A.M., L.M., G.G.N.; Funding Acquisition R.V., M.G.; Resources, R.V., M.G.; Supervision, R.V., M.G.

## Competing interests

The authors declare no competing interests.
