## [Transparent Peer Review file · Nature Communications]

Keypoint-Based Modeling Reveals Fine-Grained Body Pose Tuning in Superior Temporal Sulcus Neurons

Corresponding Author: Professor Rufin Vogels

Version 0:

Reviewer comments:

Reviewer #1

(Remarks to the Author)

In this study, the authors recorded single-neuron activity from two fMRI-identified body patches in the ventral STS (MSB and ASB) of two macaques and examined the correlation between this activity and the visual features of 720 2D static body images shown to the monkeys.

The features of the 720 images were extracted using PCA in two ways. First, the x and y coordinates of 22 key points in the 2D images were analyzed (the key point 2D model). Second, the x, y, and z coordinates of 22 key points, originally located in 3D space before being projected onto the 2D image plane, were analyzed (the 3D_VD model). Additionally, the 720 images were input into two DCNNs—AlexNet and ResNet50_rbst—and the output activations from the late-stage layers of these networks were analyzed (the AlexNet and ResNet50_rbst models, respectively).

Using cross-validated multiple linear regression, the authors assessed the correlation between neuronal activity and the 10 principal components (PCs) identified by the key point 2D and 3D_VD models, as well as the correlation between neuronal activity and the 50 PCs identified by the AlexNet and ResNet50_rbst models. To test for view invariance, the 3D_VI models derived from the 3D_VD model were also applied throughout the analysis.

The results indicated that the body patches primarily represent body poses and orientations in a view-dependent manner. The MSB and ASB showed good fits with the 2D and 3D_VD models, respectively, while both showed poor fits with the 3D_VI model. However, a few neurons in both regions exhibited some degree of view invariance.

Overall, the study is well-designed, employing modern and robust methods, and the results are both interesting and significant. I commend the authors for tackling this challenging topic. However, several points need clearer explanation to improve the manuscript, as outlined below:

1. Clarifying the implications of the 2D and 3D_VD models: The implications of the 2D and 3D_VD models differ significantly. In the main experiment, visual stimuli were presented to the monkeys as 2D images, not 3D stereoscopic images. However, both 2D and 3D_VD models were used to analyze the neural correlates. Interestingly, some neurons showed significant correlations with 3D poses in the 3D_VD model.

Throughout the manuscript, the 2D model seems to be treated almost equivalently to the 3D_VD model, which is misleading. The 2D model is based on the image (essentially a perceptual product), while the 3D_VD model involves cognitive inference. This distinction should be explicitly clarified. I recommend rewriting major sections to emphasize this point, as the ASB results are both important and substantial.

It is also important to discuss how the monkeys or neurons inferred z-values from 2D stimuli, especially given that the stimuli were presented in 2D. At least three possible sources should be considered, which are not mutually exclusive:

- Monocular stereoscopic cues: While monocular cues (e.g., occlusion, perspectives) may be present, they are likely less effective for the blurred skeletons used in this study.
- Statistical structure of the stimulus set: Recent research suggests that macaque cortical neurons can detect implicit statistical structures embedded in visual stimuli (e.g., Rajalingham et al., 2020).
- Monkeys' prior knowledge: The monkeys may have relied on visual knowledge acquired through real-life experience.

In Figure 7, the authors compared $R^2_{3D_VD}$ and R^2_{2D} . However, the role of the invisible z-dimension is intriguing. I suggest comparing $R^2_{3D_VD}$ with surrogate models where only the z-values are permuted.

2. View variance/invariance: The current manuscript's results on view invariance are not entirely convincing, and the interpretation is unclear. In the text (L237-239), the authors state that the median VII was significantly greater in ASB than in MSB, while in L333-328, they note that the 3D_VD model outperformed the 3D_VI model, indicating that few units showed view-invariant responses. It seems that VII, as used by the authors, is not a good indicator of view variance or invariance, as it merely reflects the range of neuronal responses to a particular pose shown from various views, relative to noise.

I suggest leveraging the cross-validated multiple linear regression analysis to address this topic more effectively. In lines 332-333, the authors wrote: "Although the 3D_VI model performed significantly better for the majority of units than when permuting the stimulus labels, it produced a worse fit to the data from both monkeys than the 2D and 3D_VD models (Figure 4b)." I recommend clarifying this with a rank plot of $R^2_{3D_VI}$ compared to shuffles, for both MSB and ASB, as shown in Figure 2a for R^2_{2D} and $R^2_{3D_VD}$.

There may also be an issue with Figure 4. In lines 334-335, the authors state, "The difference between the 3D_VD and 3D_VI was negatively related to the VII (Figure 4b), providing a validation of this metric of view tolerance." However, in the figure, $R^2_{3D_VI}$, $R^2_{3D_VD}$, and VII are plotted in three dimensions, including the color dimension, with VII represented by color. I suggest creating separate scatter plots of $R^2_{3D_VI}$ and $R^2_{3D_VD}$ for both MSB and ASB.

One of the most striking findings was that certain neurons in the ASB patch represented specific 3D poses with a view preference, as shown in Figure 3i-l (unit 669) and Figure 7. The frontal view preference observed in some of these ASB neurons may be related to the behavioral relevance of these poses. It would be helpful if the locations of the representative examples from Figure 3 (units 258, 626, and 669) were clearly indicated in the new figure requested above.

Additionally, the clusters in Figures 5 and 6 should indicate which representative examples from Figure 3 (units 258, 625, and 669) belong to, helping us understand whether these examples are exceptional or typical within the neuronal population.

3. Mirror symmetry: Is there any mirror symmetry in neuronal activity, similar to what has been reported in face patches? In studies of face patches, mirror symmetry emerged during the transition from full view dependence to full view invariance. Tracing mirror symmetry in this study as well would be significant and insightful.

To explore this, the authors could use their systematic stimulus set, which includes right-left symmetrical poses, along with cross-validated multiple linear regression analysis. I suggest creating a view-flipped 3D_VD model and comparing it to the original 3D_VD model and surrogate models where the z-values are shuffled. The results should be plotted separately for MSB and ASB.

4. Anatomical clustering: Since multiple single-unit activities were recorded using V-probes, the authors should assess whether there is a relationship between the clustering of neurons in PCA space (Figures 5 and 6) and the relative anatomical locations of the units. Investigating small functional structures, such as functional columns, would be significant.

5. Eye fixation: In lines 567-568, it appears that the monkeys occasionally broke eye fixation. Did they break fixation more frequently for certain poses? This could have behavioral implications, particularly for frontal pose stimuli. Addressing this may clarify the behavioral grounding of the study, which is currently underdeveloped.

(Remarks on code availability)

Reviewer #2

(Remarks to the Author)

Using fMRI-guided monkey electrophysiology in 2 body patches in STS, Raman et al. aimed at exploring the tuning to body pose. The work is extremely clear, well-written and methodologically sound. I particularly appreciated the design of the stimulus set. I have only a few questions listed below:

1. I appreciated the inclusion of AlexNet and ResNet. I agree with the authors that a simpler and interpretable model, although slightly less predictive, is preferable to block-boxes CNNs. However, I'm concerned about the fact that the key-point model might be capturing exactly the same variance of the CNNs, which would point towards a simpler, texture-matching mechanism, rather than a pose selectivity showing a bias towards socially relevant poses. I would find important to present 1) The stimulus order along the main PCs, similar to Figure S2, to show how different they look from the the PCs in Figure S2. 2) A variance partitioning analysis showing how much unique variance do the 2D and 3D models explain.
2. Similarly, I would find interesting to better quantify the difference between the 2D and 3D model, that I assume to be highly correlated. Is the unique variance between the 2 truly enough to matter?

3. The statistics is sound, and the authors deserve my compliments. I would have used the same approach. However, the chance-distribution from the permutation test seems quite liberal. What was exactly permuted? The stimulus labels? In that case, an alternative would be to permute the beta-weights across PCs, to show that the exact tuning is what's truly important.
4. I could not find the number of recording sessions, and the average yield of units per session. Did the authors combine units across multiple sessions?

Minor comments

5. Related to point 1, I do not think that acquiring extra data is necessary for it to be addressed. For this reason, I consider this as a minor point, and merely a suggestion for the authors to further strengthen the paper. I have the feeling that it would be relatively simple to produce stimuli varying along a specific PC in the key-points model whose AlexNet activations would be relatively stable (at least for individual layers). The response to such stimuli would make clear that the pose tuning goes beyond the mere spatial arrangement of the element of the scene.
6. This is somehow related to points 1 and 2. An interesting point emerging from the text is the suggestion that the tuning is biased towards poses that have a social relevance. I wonder if the authors would be able to quantify such a bias, i.e. by comparing the tuning for such poses to their frequency in nature? But I guess that monkeys spend more time in resting position, for instance, but that those are less behaviorally relevant. This result would be in line with sparse coding for behaviorally relevant information, which seems to be a recurrent topic in the visual system, e.g. selectivity to high-frequency borders in V1 and high-curvature points in V4. I am not sure if information about the likelihood of certain poses is available or if it can be derived from existing resources. If not, the authors could still spend a few words to acknowledge this aspect in the Discussion.

(Remarks on code availability)

I thank the authors for providing the notebook, all the relevant functions and the data. This is uncommon and I appreciate it. However, I might just be unfamiliar with code ocean, but it looks like I could not run the code without an account. Still, the code might benefit from more comments, especially the functions in the main notebook.

Reviewer #3

(Remarks to the Author)

In this manuscript, the authors explored the body pose selectivity of neural units recorded from body-selective patches in the middle (MSB) and anterior (ASB) superior temporal sulcus (STS) using visual stimuli of a monkey avatar displaying various body poses and views. These stimuli, derived from natural monkey poses using a deep learning algorithm, were parameterized with 2D and 3D key-point-based models (with and without view-dependency) to predict neural responses via a regression algorithm. The results showed differences in prediction performance between MSB and ASB, with MSB units exhibiting similar prediction accuracy for both 2D and 3D models, while ASB units were better predicted by the 3D model. Additionally, the view-dependent 3D model outperformed the view-independent model for ASB units. Further analysis of regression weights revealed body part specificity in MSB and a sensitivity to frontal poses in ASB units. From these results, the authors demonstrated the effectiveness of key-point-based models in understanding body pose representation in the macaque brain.

This manuscript makes a good contribution to our understanding of body pose selectivity in the STS using deep learning-generated stimuli and key-point-based parameterization of body poses. The authors employed a diverse range of experimental stimuli, allowing for the analysis of fine-grained body pose representations, which were challenging to explore with traditional methods. However, I did have concerns regarding the validity of some analytical parameter determinations, which might introduce biases in their conclusions. These issues need to be addressed to strengthen the findings and support the conclusions. Below, I provide detailed feedback and suggestions for improvement.

1. One of the strengths of the present approach is the use of diverse poses and views for the experimental stimuli (a total of 720), allowing for data-driven exploration of the characteristics and number of core dimensions (or axes) in visual body pose representations. However, the authors consistently used only 10 principal components (PCs) for the neural data analysis, which seems to underutilize the strength of their approach, though this number cannot directly be comparable to the numbers of poses, as the PCs represent axes. While they tested varying numbers of PCs to evaluate the explained variance of stimulus data (both 2D and 3D key-points, as well as CNN activations), the neural fits were only examined for CNN analysis (Figure S4). I recommend the authors test how many PCs are necessary and sufficient to explain the neural data using varying numbers of PCs with the key-point-based modeling as well. For example, it would be beneficial to confirm whether the nearly equivalent performance of 2D and 3D models in MSB units was consistently observed even when increasing the number of PCs, or this was an accidental consequence of the current parameter settings?

2. As the authors discussed in lines 323-338, a key finding of the manuscript is the view-dependent representations in the anterior STS. The authors claimed that "models that incorporated view information outperformed the view-invariant model, indicating that few if any units in MSB and ASB demonstrated view-invariant responses to different poses" (lines 336-338). However, this observation could be biased by the unit selection procedure. The authors selected units based on their selectivity for pose/view (the fourth procedure). Even if many view-invariant units exist in STS, this selection criterion may exclude them from the analysis. I recommend modifying this criterion to include units without view selectivity to test if the view-dependent model remains superior.

3. The number of analyzed units changes between different analyses, but the reason is unclear. For example, both Figure 4a

and Figure S8 used the same metric (view invariance index), but different numbers of units were analyzed. Additionally, Figure 4b does not indicate the number of units included. It would be helpful to clarify the number of units used in each analysis throughout the manuscript, and to demonstrate that the selection of units is not biased towards producing favorable results. Providing a rationale for unit selection and/or showing how robust the results are when using different selection criteria would strengthen the analysis.

4. It is unclear what proportion of the originally recorded units was ultimately analyzed. Clarifying how each selection step reduced the number of units would help readers understand the representativeness of the analyzed sample.

5. The authors compared the 2D and 3D models for ASB units, revealing their selectivity for 3D poses. However, this analysis was not performed for MSB units, making it unclear if this characteristic is specific to ASB or contributes to the better performance of the 3D model in ASB. Performing the same analysis for MSB units could clarify representational differences between MSB and ASB.

6. Some ASB units showed a high view invariance index (VII) and strong view tolerance, but the view-independent model performed poorly, leading to mixed results. It is unclear whether this is due to specific unit characteristics or a mixture of units with different properties. Clarifying this point would improve understanding of the results. Comparing the performance of the view-dependent and view-independent models directly with VII could help elucidate this issue.

7. The statement that the difference between the 3D_VD and 3D_VI models is “negatively related to the VII” (lines 334-335) needs quantitative support. I suggest providing this, as shown in Figure S8.

8. In Figure 1f, there is no explanation about the variable “t” in the figure legend, although it is explained in the Methods section. Adding an explanation in the figure legend would improve clarity.

9. In the panel for MSB of monkey T in Figure 2a, the green arrow for 3D_VD is not visible because the performances of the 2D and 3D_VD models are identical. Please adjust the figure to make this clear.

10. The permuted data in Figure 2a shows negative results, which requires explanation. Additionally, the shadowed region’s meaning (confidence interval or standard deviation?) should be clarified.

11. Line 142 refers to “Wilcoxon sign rank,” but the correct term is “Wilcoxon signed rank.” Please correct this.

(Remarks on code availability)

Although the randomness involved in the analysis seems to prevent exact replication of the results, the README file and the code provide sufficient documentation and details regarding usage, data, and explanations of variables. However, information about the required environment, such as the specific Python version and dependencies, is missing. Overall, the code includes enough information to make it possible to replicate the paper’s findings, but adding environment details would further facilitate replication.

Version 1:

Reviewer comments:

Reviewer #1

(Remarks to the Author)

The authors have responded to my comments in detail and with care, and I consider that appropriate revisions have been made.

1. Clarifying the implications of the 2D and 3D_VD models:

Based on my comments, the revised text appropriately addresses and discusses the differences between the 2D and 3D_VD models. I find the revisions to be entirely appropriate.

2. View variance/invariance:

In response to my suggestions, the authors have conducted additional analyses appropriately and explained them with illustrations in their response. Furthermore, they clearly state that such analyses may not necessarily be suitable for this study. I agree with the authors’ argument.

3. Mirror symmetry:

In response to my suggestions, the authors conducted additional analyses and referred to the results in the revised manuscript (and the supplement), adding new figures.

The authors employed appropriate and skillful approaches in conducting the additional analyses.

The results of the analyses are significant and enhance the value of the paper.

However, one important revision is necessary. The figure legend of Figure 4b is incorrect and should be revised. Additionally, I recommend a careful proofreading of the revised manuscript.

4. Anatomical clustering

The authors have appropriately conducted the additional analyses I suggested and have added a new figure in the supplement to explain their findings. I agree with their explanation.

5 . Eye fixation

The authors have appropriately conducted the additional analyses I suggested and explained them with illustrations in their response. I agree with their explanation.

(Remarks on code availability)

Reviewer #2

(Remarks to the Author)

The authors have fully addressed my concerns. Further, the additional model they have included in response to comments from the other reviewers are an excellent addition.

(Remarks on code availability)

I appreciate the addition of more comments in the code.

Reviewer #3

(Remarks to the Author)

I confirm that the authors have adequately addressed the concerns raised in my review comments, and I am fully convinced by their rebuttal. I also commend the new results they presented in response to the comments from the other reviewer. I appreciate the effort the authors put into refining their work, and I have no further concerns.

(Remarks on code availability)

Thank you for the clarification. I've confirmed that the environment details are indeed documented in environment/Dockerfile, addressing my earlier concern. I now have no further issues regarding environment setup and reproducibility.

We thank the reviewers for their excellent and supportive comments which resulted in an improved manuscript. We have responded to all comments of each reviewer. This resulted in a major revision of the main text, adding one additional main figure. We performed new data analyses, which lead to additional Suppl. Figures, a Suppl. Table and Methods. We integrated parts of the text in the Suppl. Figure legends and removed redundant text, while adding the new requested discussions and data analyses. Our responses to the reviewers' comments as well as a description of the revisions are given below in italic. The new Suppl. Figures (with legends and further description of the data) and main Figure can be found in the Suppl. Materials and manuscript, respectively.

Reviewer #1 (Remarks to the Author):

In this study, the authors recorded single-neuron activity from two fMRI-identified body patches in the ventral STS (MSB and ASB) of two macaques and examined the correlation between this activity and the visual features of 720 2D static body images shown to the monkeys.

The features of the 720 images were extracted using PCA in two ways. First, the x and y coordinates of 22 key points in the 2D images were analyzed (the key point 2D model). Second, the x, y, and z coordinates of 22 key points, originally located in 3D space before being projected onto the 2D image plane, were analyzed (the 3D_VD model). Additionally, the 720 images were input into two DCNNs—AlexNet and ResNet50_rbst—and the output activations from the late-stage layers of these networks were analyzed (the AlexNet and ResNet50_rbst models, respectively).

Using cross-validated multiple linear regression, the authors assessed the correlation between neuronal activity and the 10 principal components (PCs) identified by the key point 2D and 3D_VD models, as well as the correlation between neuronal activity and the 50 PCs identified by the AlexNet and ResNet50_rbst models. To test for view invariance, the 3D_VI models derived from the 3D_VD model were also applied throughout the analysis.

The results indicated that the body patches primarily represent body poses and orientations in a view-dependent manner. The MSB and ASB showed good predictive performances with the 2D and 3D_VD models, respectively, while both showed poor predictive performances with the 3D_VI model. However, a few neurons in both regions exhibited some degree of view invariance.

Overall, the study is well-designed, employing modern and robust methods, and the results are both

interesting and significant. I commend the authors for tackling this challenging topic. However, several points need clearer explanation to improve the manuscript, as outlined below:

1. Clarifying the implications of the 2D and 3D_VD models: The implications of the 2D and 3D_VD models differ significantly. In the main experiment, visual stimuli were presented to the monkeys as 2D images, not 3D stereoscopic images. However, both 2D and 3D_VD models were used to analyze the neural correlates. Interestingly, some neurons showed significant correlations with 3D poses in the 3D_VD model.

Throughout the manuscript, the 2D model seems to be treated almost equivalently to the 3D_VD model, which is misleading. The 2D model is based on the image (essentially a perceptual product), while the 3D_VD model involves cognitive inference. This distinction should be explicitly clarified. I recommend rewriting major sections to emphasize this point, as the ASB results are both important and substantial.

Reply: The difference between the 3D_VD and 2D model concerns the inclusion of the z-values (in-depth coordinates) of the key points, which indeed are not present in the 2D image of the avatar, but need to be computed. However, several monocular depth cues, and other factors, as discussed below, can explain the superior predictive performances for 3D_VD model compared to the 2D model. We have pointed out now this distinction between 2D and 3D_VD model in the Introduction (page 3), Results (page 6; page 16) and have discussed this point in the Discussion (page 24; see below).

It is also important to discuss how the monkeys or neurons inferred z-values from 2D stimuli, especially given that the stimuli were presented in 2D. At least three possible sources should be considered, which are not mutually exclusive:

- Monocular stereoscopic cues: While monocular cues (e.g., occlusion, perspectives) may be present, they are likely less effective for the blurred skeletons used in this study.
- Statistical structure of the stimulus set: Recent research suggests that macaque cortical neurons can detect implicit statistical structures embedded in visual stimuli (e.g., Rajalingham et al., 2020).
- Monkeys' prior knowledge: The monkeys may have relied on visual knowledge acquired through real-life experience.

Reply. The 3D_VD model indeed performed better than the 2D model in ASB and this for 2D images. As we described before, several units for which the 3D_VD model was superior to the 2D model preferred axes that described a transition from a frontal to a rear view of a monkey, or vice versa (Figure 7; Figure S14). The z coordinates of the head and trunk might be useful to differentiate these

views. The computation of the relative 3D position of the body parts can be based on relative depth information, which can be provided by monocular static, pictorial depth cues. Note that our stimuli do not consist of blurred skeletons, but sharp images of a monkey avatar in which pictorial depth cues are present. Also, visual knowledge about body structure, e.g. acquired through real-life experience, may underlie the inference of relative depth of body parts. As noted by the reviewer, the neurons might be sensitive to the inherent statistical structure, related to the 3D configuration of body parts, in 2D images of body poses. The origin(s) of the difference between the 3D_VD and 2D model requires further experimental work. We have discussed this now in the Discussion (page 24), adding references to previous studies showing the encoding of pictorial depth cues and statistical structure in images in IT.

In Figure 7, the authors compared $R^2_{3D_VD}$ and R^2_{2D} . However, the role of the invisible z-dimension is intriguing. I suggest comparing $R^2_{3D_VD}$ with surrogate models where only the z-values are permuted.

Reply: we performed the analysis suggested by the reviewer and the result is shown below in Figure Rev1. We shuffled the z values, performed PCA followed by multiple linear regression with cross-validation using the 10 PCs as predictors. In both regions, the R^2 for the shuffled z values are for most units below the original, unpermuted data for the 3D_VD model, especially for ASB. This appears to show that the z values contribute to the predictions of the 3D_VD model. However, we believe that this approach is problematic because shuffling the z values affect the PC scores of the stimuli which will indirectly affect the x and y values of the stimuli since each PC is a linear combination of the x, y and z values. Thus, it is to be expected that the predictive performances will be worse. Because we believe that shuffling z-scores are problematic we prefer not to add this analysis to the paper and keep the original, direct comparison between the 2D and 3D_VD models. The latter has now been extended by comparing the predictive performances of the 3D_VD and 2D model for different number of PCs (Figure S5): the better predictive performance for the 3D_VD compared to the 2D model was

present and significant for ASB of each monkey for all examined numbers of PCs (range 10-30 PCs).

Figure Rev1: Comparison of the original 3D_VD R^2 (red) and 3D_VD model R^2 with shuffled z values (3D_VD_SR; blue). The blue band corresponds to the percentile 2.5 and 97.5 of the R^2 for the shuffled z -score model (100 permutations of z values). Left: MSB; right: ASB. Top: data of monkey G; Bottom: data of monkey T.

2. View variance/invariance: The current manuscript's results on view invariance are not entirely convincing, and the interpretation is unclear. In the text (L237-239), the authors state that the median VII was significantly greater in ASB than in MSB, while in L333-328, they note that the 3D_VD model outperformed the 3D_VI model, indicating that few units showed view-invariant responses. It seems that VII, as used by the authors, is not a good indicator of view variance or invariance, as it merely reflects the range of neuronal responses to a particular pose shown from various views, relative to noise.

Reply: Note the 3D_VI implements a strict view-invariance, i.e. equal response for 8 views for all 45 poses, while the VII assesses the viewpoint tolerance of the model for the best pose.

We still believe that the VII is a valid (although not perfect) metric of the degree of viewpoint

tolerance of the fitted model. This is supported by new analyses in which we show that the VII, computed using the 3D_VD model, correlates significantly with the predictive performance (R^2) of the 3D_VI model (new Figure S19b) in both regions and monkeys: the higher the 3D_VI model predictive performance, the higher the VII as expected when the latter corresponds to the degree of viewpoint tolerance. Furthermore, the VII correlates negatively with the difference in predictive performance between the 3D_VD and 3D_VI model (new Figure S19c), as expected when units for which the 3D_VD performs better than the 3D_VI model have a higher view-dependency, i.e. a lower VII. Thus, we prefer to keep the VII.

I suggest leveraging the cross-validated multiple linear regression analysis to address this topic more effectively. In lines 332-333, the authors wrote: "Although the 3D_VI model performed significantly better for the majority of units than when permuting the stimulus labels, it produced a worse fit to the data from both monkeys than the 2D and 3D_VD models (Figure 4b)." I recommend clarifying this with a rank plot of $R^2_{3D_VI}$ compared to shuffles, for both MSB and ASB, as shown in Figure 2a for R^2_{2D} and $R^2_{3D_VD}$.

Reply: We have now presented a rank plot of the reliability-normalized R^2 for the 3D_VI and the null distribution obtained when shuffling stimulus labels for both MSB and ASB of both monkeys in the new Figure S15c. It shows that for the majority of units the 3D_VI model does perform better than the shuffled stimulus label model in both regions and monkeys. However, the median predictive performance is much lower than those obtained for models that keep viewpoint information, as we showed before (see new Figure 4b). For a small minority of ASB units, the predictive performances for the 3D_VD and 3D_VI model are similar (close or on the diagonal in Figure 4B), which suggests that only for a minority of ASB units there is a high degree of viewpoint tolerance. We have now stated throughout the manuscript that the predictive performances of the 3D_VI model was on average poor compared to the view-dependent models.

There may also be an issue with Figure 4. In lines 334-335, the authors state, "The difference between the 3D_VD and 3D_VI was negatively related to the VII (Figure 4b), providing a validation of this metric of view tolerance." However, in the figure, $R^2_{3D_VI}$, $R^2_{3D_VD}$, and VII are plotted in three dimensions, including the color dimension, with VII represented by color. I suggest creating separate scatter plots of $R^2_{3D_VI}$ and $R^2_{3D_VD}$ for both MSB and ASB.

Reply: we have added the requested plots relating $R^2_{3D_VI}$ and $R^2_{3D_VD}$ for MSB and ASB of each monkey (Figure 4b). In addition, we have provided scatterplots of $R^2_{3D_VD}$ and VII (new Figure S19a), and of $R^2_{3D_VI}$ and VII (new Figure S19b), with the correlations between these variables. In addition, we have added a scatterplot of the difference between $R^2_{3D_VD}$ and $R^2_{3D_VI}$ and VII (new Figure S19c). For both ASB and MSB, the correlation between the difference $R^2_{3D_VD} - R^2_{3D_VI}$ and VII was significantly negative, supporting our statement in the Results.

One of the most striking findings was that certain neurons in the ASB patch represented specific 3D poses with a view preference, as shown in Figure 3i-l (unit 669) and Figure 7. The frontal view preference observed in some of these ASB neurons may be related to the behavioral relevance of these poses. It would be helpful if the locations of the representative examples from Figure 3 (units 258, 626, and 669) were clearly indicated in the new figure requested above.

Reply: we have indicated the three units on the scatterplots.

Additionally, the clusters in Figures 5 and 6 should indicate which representative examples from Figure 3 (units 258, 625, and 669) belong to, helping us understand whether these examples are exceptional or typical within the neuronal population.

Reply: we have indicated the cluster of each unit. Unit 258 belongs to the cluster “none” of Figure S11, unit 625 to cluster 4 of Figure S11, and unit 669 to cluster 1 of Figure S12. We have indicated this now in the legends of these Figures.

3. Mirror symmetry: Is there any mirror symmetry in neuronal activity, similar to what has been reported in face patches? In studies of face patches, mirror symmetry emerged during the transition from full view dependence to full view invariance. Tracing mirror symmetry in this study as well would be significant and insightful.

To explore this, the authors could use their systematic stimulus set, which includes right-left symmetrical poses, along with cross-validated multiple linear regression analysis. I suggest creating a view-flipped 3D_VD model and comparing it to the original 3D_VD model and surrogate models where the z-values are shuffled. The results should be plotted separately for MSB and ASB.

Reply: We have followed this excellent suggestion. Before describing the model, we would like to point out that our image set did not include real mirror-symmetrical images, since these were derived from natural monkey poses with self-occlusion and asymmetric limb positions. Nonetheless, the profile and oblique views of a pose showed some rough mirror-symmetry and that was addressed by the view-flipped model. Following the reviewer's suggestion, we created a view-flipped (VF) model that have a view-flipped keypoint data matrix. For each pose and camera elevation, we replaced the rows of the 3D_VD keypoint coordinate matrices for the views at 315°, 270°, and 225° with those at 45°, 90°, and 135°, respectively. The views at 0° and 180° remained unchanged, giving

$$\mathbf{x}^{\text{view}=315^\circ} = \mathbf{x}^{\text{view}=45^\circ}; \mathbf{x}^{\text{view}=270^\circ} = \mathbf{x}^{\text{view}=90^\circ}; \mathbf{x}^{\text{view}=225^\circ} = \mathbf{x}^{\text{view}=135^\circ} \in \mathbb{R}^{1 \times 66}$$

As a control, we reversed this process such that the views at 45°, 90°, and 135° were replaced with those at 315°, 270°, and 225°, respectively. This yielded highly similar results (the median R^2 3D_VF were equal up to the second decimal place) and we will report data when replacing the coordinates of 315°, 270°, and 225° with those of 45°, 90°, and 135°. We performed PCA on the new matrices. The first 10 PCs for 3D_VF explained 87% of the variance, which is highly similar to the cumulative explained variance of the original, unflipped matrices for 3D_VD (88%). Cross-validated multiple regression with the first 10 PCs as predictors yielded R^2 values that were larger than those obtained when shuffling the stimulus labels for the large majority of MSB and ASB units of each monkey (new Figure S15d). Employing a complementary approach in which we kept the original PC space, but equated the PC scores of the mirroring views, followed by regression, produced highly similar predictive performances for the 3D_VF model in MSB and ASB compared to the approach in which the PCA was done for the view-flipped coordinates (Pearson correlations between both approaches: $r > 0.998$; the same was true for the 3D_VI modeling).

For MSB the median R^2 was significantly smaller for the 3D_VF model than for the 3D_VD model (new Figure 4c; Wilcoxon signed rank test G: $p = 8.13e-14$; T: $p = 1.45e-19$; medians R^2 are in the new Table S1), in line with units that respond differently to images having (rough) mirror-symmetry. However, for ASB, the opposite effect was present: the median R^2 was significantly larger for the 3D_VF model than for the 3D_VD model (new Figure 4c; Wilcoxon signed rank test G: $p = 2.89e-23$; T: $p = 2.38e-08$), which supports the presence of mirror-symmetric responses in many ASB units. The median R^2 for the 3D_VF model was significantly higher than for the 3D_VI model (new Figure 4d; Wilcoxon signed rank test G: $p = 7.37e-34$; T: $p = 3.73e-18$), indicating that, overall, ASB units did not have complete view-invariance. The difference between the 3D_VD and 3D_VF model predictive performances correlated significantly with the VII in MSB, with units showing a relatively worse 3D_VF predictive performance having a lower VII, computed using the 3D_VD model, in each monkey (new Figure S19d). No such correlation was present for ASB. The ASB units that showed a higher R^2 for the 3D_VF compared to

the 3D_VD model tended to show a smaller difference in R^2 between the 3D_VD compared to the 3D_2D model (new Figure S17), suggesting that units that show better mirror-symmetric viewpoint tuning differ from those that show a larger difference between 3D_VD and 2D models. Unlike the ASB units that showed a higher predictive performance for the 3D_VD compared to the 2D model, the ASB units with a better predictive performance for 3D_VF compared to the 3D_VD model showed preferred response axes describing a transformation from a quasi-frontal to a profile-like view or vice versa (new Figure 8; Figure S18). These results show that indeed several ASB units show mirror-symmetric viewpoint tuning, in line with what has been described in the face patch network (Freiwald, W.A. & Tsao, D.Y. Functional compartmentalization and viewpoint generalization within the macaque face-processing system. *Science* **330**, 845-851 (2010)). It also supports computational work that suggests that mirror-symmetric view point tuning is a general property of visual object processing (Farzmaidi, A., Zarco, W., Freiwald, W.A., Kriegeskorte, N. & Golan, T. Emergence of brain-like mirror-symmetric viewpoint tuning in convolutional neural networks. *Elife* **13** (2024)). We were happy to see that adding the mirror-symmetry viewpoint tuning increased the predictive performances of the key point model for ASB (although the R^2_{3D-VF} for ASB units was still significantly lower than those for the 3D_VD model of MSB units (Wilcoxon rank-sum tests; G: $p = 6.65e-07$; T: $p = 0.0001$)). We did not compare the 3D_VF model with surrogate models with shuffled z values, for the same reason as described above (main point 1). Furthermore, an increased predictive performance was observed also for a 2D_VF model when comparing it to the 2D model (median reliability-normalized R^2 in Table S1; ASB: G: 35% increase in median R^2 ; T: 19% increase in median R^2), showing that an improved predictive performance for the mirror-symmetry models is also present when 3D, depth information is absent.

We have added the description of the 3D_VF modeling to the main text (Abstract; Introduction; Results: page 18; Discussion: page 22) and Suppl. Material. The main points are presented in the Results (page 18) and we have added one main Figure (Figure 8) of the visualization of the axes of the units with a superior predictive performance for the 3D_VF model. Thus, we are thankful to the reviewer for suggesting this analysis.

4. Anatomical clustering: Since multiple single-unit activities were recorded using V-probes, the authors should assess whether there is a relationship between the clustering of neurons in PCA space (Figures 5 and 6) and the relative anatomical locations of the units. Investigating small functional structures, such as functional columns, would be significant.

*Reply: The current study did not aim to examine anatomical clustering of body (-part) selectivity, which is the topic of a different study that is currently in preparation for publication. Nonetheless, we computed for those units for which R^2 for the 3D_VD model was higher than 0.25 (the same criterium as for the plots of the axes in Figures 5-6) the pairwise cosine dissimilarity between the preferred axes (based on the 10 beta coefficients of the fitted models) and then applied multidimensional scaling to visualize the distances between those units in a two-dimensional space. Note that the 3D_VD model performed similarly to the 2D model for MSB units. We labeled the units according to the penetration they belonged to. As shown in Figure S20, there is a tendency for clustering of units from the same penetration in both regions and monkeys. The penetrations are estimated to be in a small volume of the STS in each region (range of grid positions, averaged across monkeys and regions, was 2 mm in the medial-lateral dimension and 1 mm in the anterior-posterior dimension per region), probes can bend along their trajectory, and penetrations were not orthogonal to the cortex because of the probe trajectory angles and curvature of the STS. Hence, an anatomical reconstruction of recording locations at columnar resolution is outside the scope of the present study. Nonetheless, the tendency for clustering of the preferred axes per penetration agrees with previous studies reporting anatomical clustering of stimulus preferences in IT (Fujita, I., Tanaka, K., Ito, M. & Cheng, K. Columns for visual features of objects in monkey inferotemporal cortex. *Nature* 360, 343-346 (1992); Dubois, J., de Berker, A.O. & Tsao, D.Y. Single-unit recordings in the macaque face patch system reveal limitations of fMRI MVPA. *J. Neurosci.* 35, 2791-2802 (2015). Verhoef, B.E., Vogels, R. & Janssen, P. Inferotemporal cortex subserves three-dimensional structure categorization. *Neuron* 73, 171-182 (2012)).*

We included the new Supplementary figure S 20 on anatomical clustering and relevant text in the Suppl. Materials and referred to it in the Discussion (page 23).

5. Eye fixation: In lines 567-568, it appears that the monkeys occasionally broke eye fixation. Did they break fixation more frequently for certain poses? This could have behavioral implications, particularly for frontal pose stimuli. Addressing this may clarify the behavioral grounding of the study, which is currently underdeveloped.

Reply: Indeed, the monkeys occasionally broke eye fixation, as expected, e.g. due to blinking. We analyzed the aborted image presentations for the frontal poses in each monkey and plotted the proportion of fixation breaks for each pose across recording sessions. These are shown below for G

(top) and T (bottom plots).

We observed differences in fixation breaks among the frontal poses, but these were rather idiosyncratic (Spearman rank correlation between horizontal view poses of G and T: $\rho = -0.24$, $p = 0.11$; correlation between top view poses of G and T: $\rho = -0.10$, $p = 0.50$) and for us difficult to interpret. Thus, we did not pursue this further.

Reviewer #2 (Remarks to the Author):

Using fMRI-guided monkey electrophysiology in 2 body patches in STS, Raman et al. aimed at exploring the tuning to body pose. The work is extremely clear, well-written and methodologically sound. I particularly appreciated the design of the stimulus set. I have only a few questions listed below:

1. I appreciated the inclusion of AlexNet and ResNet. I agree with the authors that a simpler and interpretable model, although slightly less predictive, is preferable to block-boxes CNNs. However, I'm concerned about the fact that the key-point model might be capturing exactly the same variance

of the CNNs, which would point towards a simpler, texture-matching mechanism, rather than a pose selectivity showing a bias towards socially relevant poses. I would find important to present 1) The stimulus order along the main PCs, similar to Figure S2, to show how different they look from the the PCs in Figure S2. 2) A variance partitioning analysis showing how much unique variance do the 2D and 3D models explain.

Reply: Regarding the first point of the reviewer, we have shown now the 10 images with the 10 highest and 10 lowest scores for each of the 10 PCs for the 2D model (new Figure S3). As expected there is some correspondence between the “eigenposes” of Figure S2 and the top 10 or bottom 10 images for those PCs, e.g. the change in viewpoint for PC2 is obvious for the top versus bottom images of that PC. However, images with high scores for a particular PC can be quite different, e.g. the different views for the top images for PC1. These differences are captured by other PCs. Thus, a particular image has scores for more than one PC. To quantify this, we computed for each image the Hoyer Sparsity as a metric for the sparseness of the distribution of the absolute (unsigned) PC scores of an image. A Hoyer Sparsity of 0 corresponds to equal scores for the 10 PCs, while a Hoyer Sparsity of 1 corresponds to an absolute score larger than 0 for only one of the 10 PCs. The figure below shows the distribution of the Hoyer Sparsity for all images, computed using the 2D PC scores. The mean Hoyer Sparsity was 0.36, well below 1, indicating that an image has scores for multiple PCs.

As suggested by the reviewer, we performed a variance partitioning analysis to assess the common and unique variance of the CNN models and the key point models. We used the “layer 5” activations of both networks since these produced a relatively high predictive performance for both models and regions (see Figure S6c). We performed the variance partitioning analysis for three keypoint models: 2D, 3D_VD, and a new model introduced in this revision following a comment of reviewer 1: 3D_VF, which is a model assuming equal responses to mirror-symmetric views of a pose (see our reply to point 3 of reviewer 1). The 3D_VF model provided a better predictive performance for ASB, compared to the other models. To perform the variance partitioning analysis, we fitted a combined model by

concatenating the principal component scores \mathbf{Z} from both a keypoint and a CNN model as input to a 10-fold cross-validated regression model with 60 predictors (10 from the keypoint (KP) model and 50 from the CNN). We then computed R^2 for this combined model. In addition, we computed the R^2 for each model separately. The unique and shared variance contributions of the two models were calculated as follows:

$$R_{KP\ unique}^2 = R_{Combined}^2 - R_{CNN}^2$$

$$R_{CNN\ unique}^2 = R_{Combined}^2 - R_{KP}^2$$

$$R_{Shared}^2 = R_{KP}^2 + R_{CNN}^2 - R_{Combined}^2$$

The results are shown in the new Figure S9. For MSB, the shared variance of the 2D/3D_VD models and the CNN models was higher than the unique variance for the CNNs and the latter was higher than the unique variance for the keypoint models. However, the keypoint models explained unique variance of the MSB responses, although it was numerically small. (The 3D_VF model produced on average a worse predictive performance than the 2D and 3D_VD models in MSB, and will not be discussed further.) This result shows that the keypoint models encode shape differences among the images, due to changes in pose and view, that are also captured by the CNNs, which is not surprising. The unique variance explained by the CNNs, much smaller than the shared variance component, likely results from responses to texture and shading, which are absent in the purely shape-based keypoint models. The small unique variance component of the keypoint model might capture shape differences not encoded by the texture-biased CNNs (CNNs are known to be texture-biased; see Geirhos, R., et al. ImageNet-trained CNNs are biased towards texture; increasing shape bias improves accuracy and robustness. arXiv 1811.12231 (2018).)

For ASB, the shared variance component and the unique variance component of the CNN was highly similar (Figure S9), in line with the relatively better predictive performances of the CNNs compared to the keypoint models for ASB, perhaps reflecting a relatively higher contribution of shading and/or texture cues to the selectivity in ASB compared to MSB, which requires confirmation in further studies. Note that this suggestion aligns with the higher contribution of shape compared to “appearance” to face selectivity in the midSTS face patch ML while the opposite is the case in the anterior face patch AM (Chang, L. & Tsao, D.Y. The Code for Facial Identity in the Primate Brain. *Cell* **169**, 1013-1028.e1014 (2017)). However, there was still a unique variance explained by the 3D_VD keypoint model, likely reflecting shape encoding not captured by the CNNs. The 3D_VF model shared a relatively large amount of variance with the CNNs, which is not surprising since it has been shown that CNNs also contain units with mirror-symmetric responses to various objects (Farzmaadi, A.,

Zarco, W., Freiwald, W.A., Kriegeskorte, N. & Golan, T. Emergence of brain-like mirror-symmetric viewpoint tuning in convolutional neural networks. *Elife* **13** (2024); also see Figure S19 for variance partitioning using AlexNet layer 6 units which have been reported in that paper to have greater mirror-symmetric viewpoint tuning). The unique variance components were less than the shared one for both CNNs and the 3D_VF model for ASB.

In sum, the variance partitioning analysis suggests that CNNs capture most, but not all, of the variance explained by the keypoint models. However, this does not reduce the value of keypoint models, because, as the reviewer points out, these models have as advantage that they are interpretable and their preferred pose/viewpoint selectivity can be visualized, which is very difficult, if not impossible, for the CNNs.

We have added the variance partitioning analysis (Results: page 9; Methods: page 34) and new Figures S9 and S16 to the paper.

2. Similarly, I would find interesting to better quantify the difference between the 2D and 3D model, that I assume to be highly correlated. Is the unique variance between the 2 truly enough to matter?

Reply: We have extended the comparison of the 2D and 3D_VD models by employing 2D and 3D_VD models with a different number of PC predictors, ranging from 10 to 30. Across the explored range of the number of PCs, the 3D_VD model showed consistently greater R^2 than the 2D model in ASB (but not MSB; new Figure S5). We have not performed additional analyses because the new 3D_VF model improved the predictive performances with respect to the 3D_VD model, and consequentially also the 2D model, in ASB. Nonetheless, the consistently higher predictive performances for the 3D_VD compared to the 2D model show a contribution of the relative depth coordinates of the keypoints to the selectivity of some of the ASB (but not) MSB units. This was especially the case for ASB units that show a preference for frontal or rear views of the monkey as shown in Figure 5. For these units, the difference between 2D and 3D models matters.

3. The statistics is sound, and the authors deserve my compliments. I would have used the same approach. However, the chance-distribution from the permutation test seems quite liberal. What was exactly permuted? The stimulus labels? In that case, an alternative would be to permute the beta-weights across PCs, to show that the exact tuning is what's truly important.

Reply: Indeed, we permuted the stimulus labels before performing the multiple regression. This is the same approach as in a recent study by Shi et al (Yuelin Shi, Dasheng Bi, Janis K. Hesse, Frank F. Lanfranchi, Shi Chen, Doris Y. Tsao. Rapid, concerted switching of the neural code in inferotemporal

cortex, BioRxiv, 2023). Now, we have implemented two other approaches. In the first one, we shuffled the stimulus labeling only for the test stimuli after fitting the model for the original labels. With this procedure, the R^2 values for the shuffled test data tended to be even lower. The second approach followed the suggestion by the reviewer: we permuted the beta weights across the 10 PCs when predicting the responses after fitting the model. Again, this produced negative R^2 for the shuffled data. We have added these alternative methods now to the Methods (page 34) and added the new Figure S15 that compares the different approaches.

4. I could not find the number of recording sessions, and the average yield of units per session. Did the authors combine units across multiple sessions?

Reply: We had 11 and 12 recording sessions in MSB of monkey G and monkey T, respectively, and 15 and 6 in ASB of G and T, respectively. Our selection criteria for responsivity, selectivity and reliability yielded 346 reliably selective MSB units (average yield per session: 10.5 units) and 544 ASB units (average yield per session: 25.9 units). The to-be-modeled units were required to be recorded from channels that were shown to be body-category selective ($BSI > 0.33$) in an independent test. For MSB this resulted in 285 modeled units (average yield of 8.6 units per session) and for ASB in 305 units (14.5 per session). We have added these data to the Methods (page 26; page 29). The recordings were acute (daily penetrations) and thus it was impossible to record from the same unit across multiple sessions.

Minor comments

5. Related to point 1, I do not think that acquiring extra data is necessary for it to be addressed. For this reason, I consider this as a minor point, and merely a suggestion for the authors to further strengthen the paper. I have the feeling that it would be relatively simple to produce stimuli varying along a specific PC in the key-points model whose AlexNet activations would be relatively stable (at least for individual layers). The response to such stimuli would make clear that the pose tuning goes beyond the mere spatial arrangement of the element of the scene.

Reply: we showed in the variance partitioning analysis that there is, although small, unique variance for the keypose models. Thus, it is expected that in the case it is possible to create monkey images that produce an equal activation in some AlexNet layers but differ in keypoint coordinates, there will be differences in neural response for (some of?) these images. We did not have the opportunity to test this prediction in the STS. Note that the 2D and 3D_VD keypoint model quantifies the spatial

arrangement of the body parts in a body. It allows us to visualize the tuning of the neurons for specific changes in the spatial arrangement of (some of) the body parts, i.e. changes in pose and view. The latter is not possible with the CNNs. The 3D_VF model goes one step further by introducing a mirror-symmetry invariance.

6. This is somehow related to points 1 and 2. An interesting point emerging from the text is the suggestion that the tuning is biased towards poses that have a social relevance. I wonder if the authors would be able to quantify such a bias, i.e. by comparing the tuning for such poses to their frequency in nature? But I guess that monkeys spend more time in resting position, for instance, but that those are less behaviorally relevant. This result would be in line with sparse coding for behaviorally relevant information, which seems to be a recurrent topic in the visual system, e.g. selectivity to high-frequency borders in V1 and high-curvature points in V4. I am not sure if information about the likelihood of certain poses is available or if it can be derived from existing resources. If not, the authors could still spend a few words to acknowledge this aspect in the Discussion.

Reply: This is an interesting suggestion. Several studies monitored the behavior of rhesus monkeys in natural, rural, and urban settings (e.g. Jaman MF, Huffman MA. The effect of urban and rural habitats and resource type on activity budgets of commensal rhesus macaques (Macaca mulatta) in Bangladesh. Primates. 2013 Jan;54(1):49-59). These studies use behavioral categories such as eating, locomotion, aggressive interaction, grooming, resting, etc. However, these categories can include multiple poses and views, as observed from the perspective of monkeys observing other monkeys. Typically, resting and eating/foraging are highest in frequency, thus one may assume that these are the most common poses viewed by monkeys. MSB units tended not to prefer sitting or eating poses, the more frequent ones observed in natural settings, which fits the conjecture of the reviewer. ASB units with a high 3D-VD model predictive performance preferred often frontal or rear views that happen to have social meaning, but this is a post-hoc observation. Furthermore, we argued that the combination of pose and viewpoint selectivity can be ecologically or socially relevant since the orientation of a particular pose towards another agent has obvious social implications. However, we hesitate to make strong claims about the frequency distributions of pose preferences of units in the STS or the regions we recorded from (see Discussion, page 23). The reason is that we cannot exclude that the distribution of the pose preference of the units results from a restricted sampling because preferences within a penetration tend to cluster (see new Figure S20). Thus, as we stated before when discussing differences in pose-view preferences between MSB and ASB, one needs to be cautious in interpreting such differences, and, by implication, the distributions of pose preferences. As a result,

we did not include this interesting suggestion by the reviewer in the Discussion, but it should be addressed in a future study using wider sampling along the STS.

Reviewer #2 (Remarks on code availability):

I thank the authors for providing the notebook, all the relevant functions and the data. This is uncommon and I appreciate it. However, I might just be unfamiliar with code ocean, but it looks like I could not run the code without an account. Still, the code might benefit from more comments, especially the functions in the main notebook.

Reply: We are not entirely sure what the reviewer interface on Code Ocean looks like, as this is new to us as well. A login might be required to run the code, or reviewers could download the 'capsule' and execute the code on their local machines. However, the capsule runs headlessly and includes all the logs for the results. The idea is that reviewers can inspect the output at a glance without needing to rerun the code (<https://help.codeocean.com/en/articles/2644233-peer-review-on-code-ocean>). Additionally, we have added more comments throughout the project for clarity.

Reviewer #3 (Remarks to the Author):

In this manuscript, the authors explored the body pose selectivity of neural units recorded from body-selective patches in the middle (MSB) and anterior (ASB) superior temporal sulcus (STS) using visual stimuli of a monkey avatar displaying various body poses and views. These stimuli, derived from natural monkey poses using a deep learning algorithm, were parameterized with 2D and 3D key-point-based models (with and without view-dependency) to predict neural responses via a regression algorithm. The results showed differences in prediction performance between MSB and ASB, with MSB units exhibiting similar prediction accuracy for both 2D and 3D models, while ASB units were better predicted by the 3D model. Additionally, the view-dependent 3D model outperformed the view-independent model for ASB units. Further analysis of regression weights revealed body part specificity in MSB and a sensitivity to frontal poses in ASB units. From these results, the authors demonstrated the effectiveness of key-point-based models in understanding

body pose representation in the macaque brain.

This manuscript makes a good contribution to our understanding of body pose selectivity in the STS using deep learning-generated stimuli and key-point-based parameterization of body poses. The authors employed a diverse range of experimental stimuli, allowing for the analysis of fine-grained body pose representations, which were challenging to explore with traditional methods. However, I did have concerns regarding the validity of some analytical parameter determinations, which might introduce biases in their conclusions. These issues need to be addressed to strengthen the findings and support the conclusions. Below, I provide detailed feedback and suggestions for improvement.

1. One of the strengths of the present approach is the use of diverse poses and views for the experimental stimuli (a total of 720), allowing for data-driven exploration of the characteristics and number of core dimensions (or axes) in visual body pose representations. However, the authors consistently used only 10 principal components (PCs) for the neural data analysis, which seems to underutilize the strength of their approach, though this number cannot directly be compared to the number of poses, as the PCs represent axes. While they tested varying numbers of PCs to evaluate the explained variance of stimulus data (both 2D and 3D key-points, as well as CNN activations), the neural fits were only examined for CNN analysis (Figure S4). I recommend the authors test how many PCs are necessary and sufficient to explain the neural data using varying numbers of PCs with the key-point-based modeling as well. For example, it would be beneficial to confirm whether the nearly equivalent performance of 2D and 3D models in MSB units was consistently observed even when increasing the number of PCs, or this was an accidental consequence of the current parameter settings?

Reply: We have now provided the predictive performance (R^2) as a function of the number of PCs (from 10 to 30 PCs) for the 2D and 3D_VD key pose model for each region and monkey in the new Figure S5. This new analysis shows that the 3D_VD key pose model outperforms significantly the 2D model irrespective of the number of PCs, while this was not the case for MSB. Thus, the difference between MSB and ASB was not a consequence of the choice of the number of PCs. We have added this to the Results, page 7)

2. As the authors discussed in lines 323-338, a key finding of the manuscript is the view-dependent representations in the anterior STS. The authors claimed that “models that incorporated view information outperformed the view-invariant model, indicating that few if any units in MSB and ASB demonstrated view-invariant responses to different poses” (lines 336-338). However, this observation could be biased by the unit selection procedure. The authors selected units based on

their selectivity for pose/view (the fourth procedure). Even if many view-invariant units exist in STS, this selection criterion may exclude them from the analysis. I recommend modifying this criterion to include units without view selectivity to test if the view-dependent model remains superior.

Reply: we did not test specifically for viewpoint selectivity when selecting units: we selected units that show a different response to the 720 stimuli. These stimuli included images of 45 different poses, each from 16 views. Thus, they varied not only in viewpoint but, importantly, also in pose. Even if a pose-selective unit would have been view-invariant, it should have been selected by our test since the responses would differ among the 45 poses, because of its pose selectivity. Thus, we do not see how applying the ANOVA test would not have selected view-invariant units that are also pose-selective. Units that would have responded to all images equally well would not have been selected, but such units were not of interest to us in the present study – these cannot be modeled-. Furthermore, our results on viewpoint selectivity are in line with a previous study of the ASB region that did not report single units with complete view-invariance (Kumar, S., Popivanov, I.D. & Vogels, R. Transformation of Visual Representations Across Ventral Stream Body-selective Patches. Cereb Cortex, 1-15 (2017); see their Figure 5).

3. The number of analyzed units changes between different analyses, but the reason is unclear. For example, both Figure 4a and Figure S8 used the same metric (view invariance index), but different numbers of units were analyzed. Additionally, Figure 4b does not indicate the number of units included. It would be helpful to clarify the number of units used in each analysis throughout the manuscript, and to demonstrate that the selection of units is not biased towards producing favorable results. Providing a rationale for unit selection and/or showing how robust the results are when using different selection criteria would strengthen the analysis.

Reply: The comparison of the predictive performance between models was performed for all units that obeyed our selection criteria. The comparison of the View Selectivity Index (VII) between MSB and ASB units (Figure 4a) was performed for those units for which the (reliability-unnormalized) R^2 was larger than 0.25. The reason is that the VII is computed using the model responses and thus we employed only units with a good model fit in both regions. However, we have included now also the results of this MSB – ASB comparison for VII for all units, independent of their goodness of fit (Figure S10). When including all units, ASB still had a significantly greater median VII than MSB for both the 2D and 3D_VI models and in both animals (Figure S10). Thus, this difference did not result from selecting only units with good model predictive performances. The visualization of the axes was done

for all units but we show in the Figures those units with a fit larger than 0.25. Again, the reason is that axes will be unreliable when the predictive performance is low. We have now made this more explicit and indicate for each figure the number of units included in the analysis and how these were selected.

4. It is unclear what proportion of the originally recorded units was ultimately analyzed. Clarifying how each selection step reduced the number of units would help readers understand the representativeness of the analyzed sample.

Reply: Our selection criteria for responsivity, selectivity, and reliability, kept 53% of the sorted units, averaged across monkeys, for MSB, and (also) 53% for ASB. These criteria were applied simultaneously (using an “AND” operation) and not successively. The implementation of these criteria was necessary because the pose and/or viewpoint selectivity can only be properly modeled for units having a reliable selective response to begin with, i.e. for which the response strength differs reliably amongst the poses and/or views. This yielded 346 reliable viewpoint and/or pose selective MSB units and 544 ASB units. The to-be-modeled units were required to be recorded from channels that were body-category selective ($BSI > 0.33$) in an independent test, which was the case for 76% and 91% of the reliably pose-and/or view-selective MSB units of monkey G and monkey T, respectively, and for 49% and 78% of the reliably selective ASB units of G and T, respectively. Thus, for MSB, this resulted in 285 modeled units, and for ASB in 305 units. We employed the body-category selectivity criterium to avoid including units from face-category selective channels and channels for which average object and face versus body responses were similar. We have included these data and rationales now in the Methods section (page 29).

5. The authors compared the 2D and 3D models for ASB units, revealing their selectivity for 3D poses. However, this analysis was not performed for MSB units, making it unclear if this characteristic is specific to ASB or contributes to the better performance of the 3D model in ASB. Performing the same analysis for MSB units could clarify representational differences between MSB and ASB.

Reply: as we described before, the median R^2 for the 2D and 3D_VD models did not differ significantly. Also, as can be seen in the new Figure S17, the difference of the predictive performances between the 2D and 3D_VD model is centered on zero for MSB, unlike for ASB, with very few units showing a sizable difference between 3D_VD and 2D in MSB. Because only very few MSB units showed a sizable difference between the 2D and 3D_VD models (Figure S17), we did not perform this analysis for MSB. A clear representational difference between MSB and ASB was revealed for a new model, 3D_VF,

which equates responses to mirror-symmetrical profiles and oblique views. This new model was introduced based on a comment from reviewer 1 (see main point 3 of reviewer 1 for more details). As shown in the same Figure S17, the 3D_VF model provides a worse predictive performance compared to VD_3D (and 2D; not shown in that figure but median predictive performances are in Table S1) in MSB but the opposite was the case for ASB. These new results have been described in the paper now (Results: page 18; new Figure 8) and Suppl. Material.

6. Some ASB units showed a high view invariance index (VII) and strong view tolerance, but the view-independent model performed poorly, leading to mixed results. It is unclear whether this is due to specific unit characteristics or a mixture of units with different properties. Clarifying this point would improve understanding of the results. Comparing the performance of the view-dependent and view-independent models directly with VII could help elucidate this issue.

Reply: Note the 3D_VI implements a strict view-invariance, i.e. equal response for 8 views for all 45 poses, while the VII assesses the viewpoint tolerance of the model for the best pose. As suggested by the reviewer, we have compared now the predictive performances of the view-dependent and view-independent models with VII. As expected, we observed no significant correlation between VII and the performance of the 3D_VD model in either region of monkey G, but for monkey T, we found significant correlations between VII and the predictive performances of the 3D_VD model, with higher VIIs for better predictive performances. Note that the VII is computed using the 3D_VD model, making this correlation between predictive performances and VII for that model difficult to interpret. (Because of this correlation, we used the R^2 for the 3D_VD model as a control variable (partial correlation) in analyses where we correlated VII, computed using that model, and the other variables (new Figure S19)). The VII correlated positively with the 3D_VI predictive performances for each monkey and region, which is expected when the VII captures the degree of viewpoint tolerance (Figure S19). The VII correlated significantly, and negatively, also with the difference in 3D_VD – 3D-VI predictive performances (Figure S19): the higher the VII the relatively less bad were the predictive performances for the 3D_VI model, and this in both regions.

7. The statement that the difference between the 3D_VD and 3D_VI models is “negatively related to the VII” (lines 334-335) needs quantitative support. I suggest providing this, as shown in Figure S8.

Reply: We have now provided this Figure (new Figure S19c) and the negative correlations were significant in each monkey and region.

8. In Figure 1f, there is no explanation about the variable “t” in the figure legend, although it is explained in the Methods section. Adding an explanation in the figure legend would improve clarity.

Reply: we have now provided such explanation in the Figure legend of Figure 1.

9. In the panel for MSB of monkey T in Figure 2a, the green arrow for 3D_VD is not visible because the performances of the 2D and 3D_VD models are identical. Please adjust the figure to make this clear.

Reply: we have now adjusted the figure to make this clear.

10. The permuted data in Figure 2a shows negative results, which requires explanation. Additionally, the shadowed region’s meaning (confidence interval or standard deviation?) should be clarified.

Reply: The coefficient of determination will be negative when the model fits the data worse than just taking the mean of the dependent variable. This can happen when permuting the stimulus labels because the original relationship between the predictors and the dependent variable is destroyed (also see Yuelin Shi, Dasheng Bi, Janis K. Hesse, Frank F. Lanfranchi, Shi Chen, Doris Y. Tsao, Rapid, concerted switching of the neural code in inferotemporal cortex, BioRxiv, 2023) for negative R^2 when shuffling stimulus labels). We have added this explanation to the Methods (page 34). The shadowed regions of the figures show the 2.5 and 97.5 percentiles of these “null distributions”. We have added this explanation to the captions.

11. Line 142 refers to “Wilcoxon sign rank,” but the correct term is “Wilcoxon signed rank.” Please correct this.

Reply: thank you for spotting this typo. We have corrected it.

Reviewer #3 (Remarks on code availability):

Although the randomness involved in the analysis seems to prevent exact replication of the results, the README file and the code provide sufficient documentation and details regarding usage, data, and explanations of variables. However, information about the required environment, such as the specific Python version and dependencies, is missing. Overall, the code includes enough information to make it possible to replicate the paper’s findings, but adding environment details would further

facilitate replication.

Reply: The Code Ocean repository contains details about the environment in environment/Dockerfile and .codeocean/environment.json. We hope the reviewer can access it.

Point-by-point reply

We thank the three reviewers for their positive evaluation of our work. Reviewer 1 had one remaining comment:

“However, one important revision is necessary. The figure legend of Figure 4b is incorrect and should be revised.

Additionally, I recommend a careful proofreading of the revised manuscript.”

Reply: We thank the reviewer for identifying this copying error. We have corrected it and carefully proofread the manuscript, making additional corrections where necessary.